# ASSESSING LARGE LANGUAGE MODELS FOR VALID AND CORRECT CODE REASONING

## ABSTRACT

Frontier large language models (LLMs) consider reasoning as first-class citizens: they learn to refine their reasoning process and try different strategies during training. Thereby, when prompted, can think through problems and respond better with proper reasoning. For programming tasks, this makes code reasoning a must. In this paper, we propose the task of *Code Execution Simulation (CES)* as a proxy for evaluating the code reasoning capabilities of LLMs. CES defines the notions of *valid* or *invalid* reasoning process, which enables it to *promptly* (1) determine where the execution simulation diverges from ground truth for incorrect output predictions (essential to understanding limitations of LLMs in code reasoning) and (2) identify suspiciously correct output predictions (essential to understanding reasoning shortcuts, hallucinations, or potential data leakage). In addition to evaluating LLMs' execution reasoning on a program with a single test, CES measures their reasoning consistency across tests with the same or different prime path coverage. This enables it to evaluate the code reasoning of LLMs in a spectrum: *strong*, *weak*, and *random*. Our results show that LLMs, to a great extent (83.32%), follow a valid reasoning process (results in 30.79% correct and 51.53% incorrect output predictions). However, their reasoning is mostly random (55.59%) or weak (41.69%), which explains their weakness in programming tasks that require flow- or path-sensitive program analysis to succeed.

## 1 INTRODUCTION

Large Language Models (LLMs) have shown emerging abilities in code/test synthesis, bug/vulnerability detection, code translation, and program repair (Roziere et al., 2023; Zhu et al., 2024; Achiam et al., 2023; Lozhkov et al., 2024; Reid et al., 2024; Yang et al., 2024; Mishra et al., 2024). The extent to which they can reason about code execution is still under investigation. CRUXEVAL (Gu et al., 2024) and CODEMIND (Liu et al., 2024a) proposed the output prediction task for given inputs to evaluate LLMs for code execution reasoning. REVAL (Chen et al., 2024) took one step further and evaluated LLMs using four runtime behavior prediction tasks: for given inputs and a statement in the program, predict (1) if the statement is covered during execution, (2) variable values after the execution of it, (3) the next statement executed after it, and (4) final output.

These techniques lack the following essential features: **Flow Sensitivity.** Assessing LLMs for following the correct execution path from start to end is essential for code reasoning. CRUXEVAL and CODEMIND only concern output prediction without evaluating intermediate program states. REVAL evaluates runtime behavior prediction of *a subset of statements* in programs, not all, due to computational overhead inherent in their design. It also prompts an LLM separately per individual statement without strategies to combine all predictions for the entire program. **Diagnosis.** None of these techniques can *promptly* and *reliably* determine (1) where the LLM's execution reasoning diverges from the ground truth and results in *incorrect output prediction*, or (2) flags *suspiciously correct output predictions*. The former is essential to understanding the limitations of LLMs in code reasoning and improving the next generation of Code LLMs. The latter can reveal data contamination, hallucinations, and reasoning shortcut cases, which are important trustworthiness concerns (Zhang et al., 2023; Shi et al.; Ding et al., 2024a). **Reasoning Consistency.** Prior techniques evaluate the reasoning of LLMs per single test, failing to study the consistency of reasoning across multiple tests with potentially different coverage. Investigating such consistency can reveal the strength of inductive reasoning in LLMs: a model that correctly reasons about execution across tests with different coverage is unlikely to succeed by chance (*Strong Reasoning*), and a model that correctly reasons about the execution across tests with the same coverage, but not those with difference coverage has a *Weak Reasoning* abilities. Otherwise, the reasoning of the model can be considered random.

We propose *Code Execution Simulation (CES)* task for assessing LLMs in code reasoning. CES unifies output prediction and intermediate program state predictions into one prompt. Asking LLM to predict all intermediate program states and output makes the task complex and can confuse it,

Figure 1: A motivating example showcasing CES reasoning (c) in root causing the incorrect output prediction by GPT-4 Turbo for HumanEval/37 problem (b), compared to REVAL reasoning (a)

especially since this requires reasoning across multiple statements considering execution flow (Al-lamanis et al.; Sabbatella et al., 2024; Chang et al., 2024). We alleviate the issue by (1) prompting the model to predict only essential decision point values (loop variables, loop iterables, predicates, conditions, and branches) and (2) instructing the task with adaptive in-context examples (using static analysis to adjust the in-context example based on the program). CES is *flow-sensitive* (evaluates the execution reasoning of a program as a whole), *scalable* (need to prompt model per program and test), and *diagnostic* (identifies execution simulation divergence points for incorrect output predictions AND rule out suspicious correct output predictions).

Extensive empirical evaluation of CES on two API-access (GPT-4 Turbo and Gemini-1.5 Pro) and eleven open-access LLMs (different sizes and training strategies of DeepSeekCoder, CodeLlama, MagiCoder, StarCoder2, and SemCoder) demonstrates that LLMs, in general, can follow a valid reasoning process, resulting in 30.79% correct and 51.53% incorrect output predictions. However, their reasoning is mostly random (55.59%) or weak (41.69%). Our experiments show slight agree-ment between CES and three other programming tasks that heavily rely on program understanding and analysis (bug detection, localization, and repair), even for SemCoder that is pre-trained on ex-ecution data. This confirms why LLMs have yet to generalize to programming tasks, specifically in the real world and beyond benchmarks. CES also categorizes the root causes for incorrect or suspiciously correct output predictions, which could be valuable to understanding the limitations of LLMs in code reasoning and designing the next generation of execution-aware Code LLMs. Our artifacts are publicly available (Authors, 2024).

## 2 MOTIVATING EXAMPLES

Understanding where the model's execution simulation diverges from real execution can reveal im-portant facts about LLMs' code understanding. Can existing code execution reasoning approaches diagnose the inception of such divergence? To answer this question, consider the example in Fig-ure 1-b, which shows the code and corresponding test for the HumanEval/37 problem. CODEMIND, REVAL, and CES unanimously show that GPT-4 Turbo *cannot* correctly predict the output of this program for the given inputs. CODEMIND provides no additional information on where the model lost the execution track, producing incorrect output prediction. REVAL selects statements 7, 9, and 10 as representative statements for code reasoning. For each statement, it individually prompts GPT-4 Turbo to predict if the statement is covered in the execution (prompt1), what are the variable values after the execution (prompt2), and what is the next statement to be executed (prompt3). For the last statement, it also asks the model to predict the final output value (prompt4). Figure 1-a summarizes the GPT-4 Turbo response to REVAL's 10 prompts.

GPT-4 Turbo correctly predicted prompt1 for all selected statements. When it comes to loops, REVAL's ground truth values for predicting program states (prompt2) and the next statement to be executed (prompt3) consider the first loop iteration. Looking at the results of prompt2, we can see that GPT-4 Turbo correctly predicts the value of `ans` in the first iteration, but the predicted value for this variable in statements 9 and 10, and hence the output, are different from the ground truth values. The model's response to the next statement execution prediction (prompt3) does not match the ground truths in any statement.

This example highlights the issues of individual statement reasoning of REVAL and inconsistencies that prevent diagnosis of execution simulation divergence: the reasoning response corresponding to statement 7 (highlighted in purple) suggests that LLM terminates the loop after the first iteration (it predicts statement 8 to execute next). When prompting the model to predict runtime properties

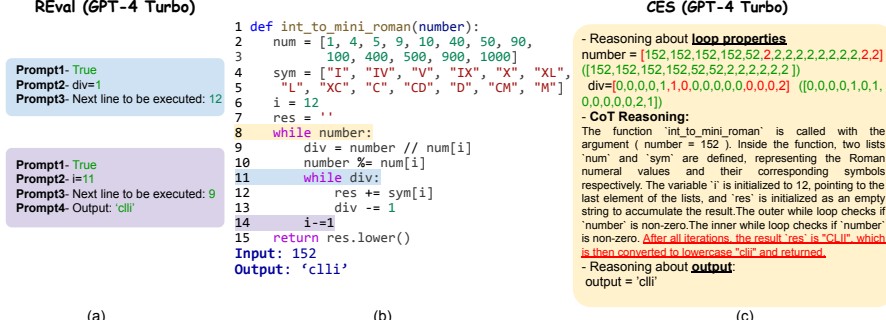

Figure 2: A motivating example showcasing CES reasoning (c) in identifying suspiciously correct prediction of GPT-4 Turbo for the HumanEval/156 problem (b), compared to REVAL reasoning (a)

of statement 9, the response (highlighted in blue) suggests that LLM does iterate over the loop and populates `ans` with more items (inconsistency 1). Looking at the model's response to reasoning about statement 10 (highlighted in green), we can see that there is also an inconsistency in predicting the value of `ans` and predicting output, which are the same thing (inconsistency 2). Given all these logical inconsistencies, one cannot reliably identify execution simulation divergence using REVAL.

CES prompts LLMs to predict values of loop variables (`e` and `o`), loop iterable (`zip(evens, odds)`), predicate (`len(evens)>len(odds)`), branch (`if` statement), and output (`ans`). Figure 1-c shows the response of GPT-4 Turbo to CES's prompt. We can clearly see that the model simulates the execution from the program's start to the end, and predicted values, even if incorrect, are consistent throughout the program. These results suggest that the divergence from ground truth is due to misprediction of the loop variable `o` at statement 6. Although the simulation correctly follows the control and data flow of the program, the misprediction of `o` propagates and results in incorrect output prediction. This diagnosis is possible by prompting GPT-4 Turbo *once*, compared to 10 prompts in REVAL that cannot provide reliable insights.

In addition to identifying simulation divergence, it is also important to *rule out* suspiciously correct output predictions. Such cases can happen due to data contamination (the expected output of the program for given inputs has been seen during training), hallucinations (correct prediction based on previously incorrect ones), or shortcuts (predicting the code logic based on the function name not understanding the code). As we will show in this paper, this phenomenon is common, even in the best LLMs such as GPT-4 Turbo and Gemini-1.5 Pro (§5.4).

Figure 2 demonstrates the ability of CES in detecting such case when prompting GPT-4 Turbo for reasoning about HumanEval/156 problem, compared to REVAL and CODEMIND. All three approaches show that GPT-4 Turbo correctly predicts the output for the given input. GPT-4 Turbo correctly predicts prompt1, prompt2, and prompt3, thereby, REVAL marks its reasoning (incrementally) consistent. On the other hand, GPT-4 Turbo fails to correctly predict the value of loop variable `numbers` in Line 8, starting from iteration 6. This impacts the prediction related to the inner loop, resulting in incorrect prediction of values for `div` at the same iteration. In real execution, these incorrect predictions would propagate to an incorrect output prediction. However, the predicted output matches the ground truth. CES marks this reasoning process invalid and discards the suspiciously correct output prediction. Further manual investigation shows that even the step-by-step natural language reasoning of the model for this case is incomplete (it only explains a high level of code without considering program states and variable values) and incorrect (finally, it incorrectly predicts the output to be "clii" rather than "clli"). Thereby, CES correctly identified and discarded it.

## 3 CODE EXECUTION SIMULATION (CES)

A program $P = \{s_i | s_i \in S_{assign} \cup S_{loop} \cup S_{condition} \cup S_{return}\}$ is a set of statements that can assign a value to a variable ($S_{assign}$), introduce recursions in the logic ($S_{loop}$), cause branches in the control flow ($S_{condition}$), or terminate the program execution ($S_{return}$). Given input(s) $I$ to $P$, CES evaluates LLMs in predicting properties related to $S_{loop}$, $S_{condition}$, and $S_{return}$ statements. These statements identify the start or end of basic blocks in the program and can capture mispredictions in the assignment statements inside the block. Furthermore, asking a model to predict all intermediate variable values can make the task complex, preventing proper responses to evaluate them (Liu et al., 2024b). In this section, we first define these properties (§3.1) and explain how the CES prompts models and evaluates their abilities in simulation execution (§3.2). We also define the notions of

valid or invalid reasoning process (§3.3) and how this enables CES to diagnose suspiciously correct output prediction or determine where the simulation diverges from execution (§3.4).

## 3.1 PROGRAM PROPERTIES DEFINITIONS

**Definition 1. Loop Properties.** A program may contain $m$ $(m \geq 0)$ loop statements. A loop statement $l_j \in S_{loop} = \{l_1, \ldots, l_m\}$ consists of two main components: loop variable $(V_{l_j})$ and loop iterable $(I_{l_j})$[1]. The loop variable keeps track of the iterations, and the loop iterable defines the values or orders of the loop variable. In the example of Figure 1-b, the statement in Line 6 is a loop statement. The loop variables are `e` and `o`, and the loop iterable is `zip(evens,odds)`. For each $l_j$ in a given program $P$ and concerning input(s) $I$, CES will ask the LLM to predict the values of all $V_{l_j}$s. The loop iterable can be a compound, i.e., consists of multiple variables or API calls. As a result, CES asks the LLM to predict values of all sub-components. The rationale here is to correctly identify the root cause for simulation divergence. In the example of Figure 1-b, the LLM may diverge from ground truth by mispredicting the values of `evens` or `odds`, or it may fail to understand the logic of `zip` API in Python library, mispredicting the return value of it even with correct values of `evens` and `odds`.

**Definition 2. Condition Properties.** A program may contain $n$ $(n \geq 0)$ conditional statements. A conditional statement $c_j \in S_{condition} = \{c_1, \ldots, c_n\}$ represents a branch $B_{c_j}$ in the control flow, which may be taken or not during execution, depending on the value of the conditional predicate $P_{c_j}$. A predicate can be a compound, i.e., consisting of multiple sub-predicates connected by logical operators. For example, a conditional statement `if (x > 0 && y < 0)` consists of two sub-predicates, `x > 0` and `y < 0`. Compared to related research (Chen et al., 2024; Tufano et al., 2023) that evaluate LLMs only for predicting the branches to be taken or not, CES performs a finer-granularity assessment, asking LLMs to predict all sub-predicates, the predicate, and the branch. A misprediction at each could indicate a specific limitation of LLM in code reasoning: an LLM that correctly predicts sub-predicates but not the predicate may struggle with reasoning about complex logical expressions. An LLM that can correctly predict sub-predicates and predicate, but not the branch, struggles in understanding program construct semantics. As we will explain later in this section, this level of granularity also enables CES to identify invalid reasoning.

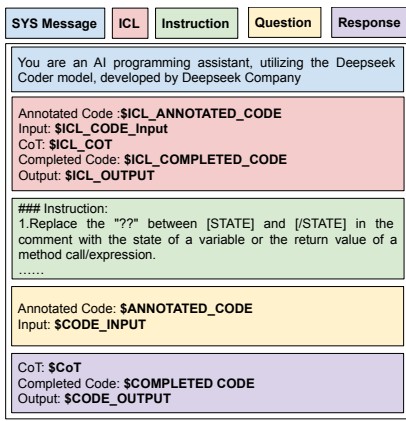

Figure 3: Prompt template used in CES

**Definition 3. Return Properties.** A program may contain $k$ $(k \geq 0)$ conditional statements. A return statement $r_j \in S_{return} = \{r_1, \ldots, r_k\}$ defines the output of the program (a value to be returned or message to be logged), $O_{r_j}$, once the execution terminates. A return statement can also be compound; in such cases, CES breaks it down into the sub-components and evaluates the LLM in predicting the values per each.

## 3.2 PROMPTING AND METRICS

Figure 3 shows the prompt template in CES. Regardless of the number of loops, conditions, and return statements, and whether the loops or conditions are nested, CES asks the model to predict all properties during the execution simulation of given inputs. This enables a flow-sensitive assessment of the models in code execution simulation, showing where the simulation starts to converge from real execution. Figure 4 demonstrates the annotated code in the prompt used to evaluate GPT-4 Turbo in the illustrative example

```python
def sort_even(l: list):
    evens = l[::2]
    odds = l[1::2]
    evens.sort()
    ans = []
    for e, o in zip(evens, odds): ## [STATE]e=??[/STATE]
[STATE]o=??[/STATE][STATE]evens=??[/STATE][STATE]odds=??
[/STATE][STATE]zip(evens,odds)=?? [/STATE]
        ans.extend([e, o])
    if len(evens) > len(odds): ## [CONDITION](len(evens) >
len(odds))=??[/CONDITION][BRANCH]taken=??[/BRANCH]
        ans.append(evens[-1])
    return ans
```

Figure 4: Annotated code in CES's prompt

of Figure 1. CES leverages in-context learning (Brown et al., 2020) to introduce code execution simulation task to LLMs.

---

[1] Note that loop iterable may not explicitly be specified in the loop statement, e.g., highlighted `while` statements in Figure 2-b only have loop variables.

When in-context examples closely resemble the problem, performance improves significantly due to the model's generalization from familiar patterns (Ye et al., 2023; Zhang et al., 2022). As a result, CES constructs a pool of examples reflecting different combinations of programming constructs (details in §A.1), e.g., nested loops or conditional statements. When constructing the prompt, it performs a lightweight static analysis on $P$ to find the most relevant in-context example from the pool. It also prompts LLMs with implicit Chain-of-Thoughts (CoT) (Wei et al., 2022). After receiving the response of model $M$ for simulating the execution of program $P$ under inputs $I$, CES compares the ground truth with the predicted values for properties of individual statement in $l_j \in S_{loop}$, $c_j \in S_{condition}$, and $r_j \in S_{return}$ as below:

$$CES(M, P, I, X_{y_j}) = [\![\sum_{w=1}^{z} [\![M(P, I, X_{y_{j_w}}) = GT(P, I, X_{y_{j_w}})]\!] = z]\!] \tag{1}$$

where $X_{y_j}$ represents individual $V_{i_j}$, $I_{l_j}$, $B_{c_j}$, $P_{c_j}$, and $O_{r_j}$ properties[2]. $X_{y_j}$ can be compound, consisting of multiple sub-components $X_{y_{j_w}} = \{X_{y_{j_1}}, \ldots, X_{y_{j_z}}\}$. As a result, Equation 1 evaluates whether model's prediction and ground truth values for sub-components of property $X_{y_j}$ match or not. $CES(M, P, I, X_{y_j})$ is 1 only if the model correctly predicts all sub-components of $X_{y_j}$.

## 3.3 Determining the Validity of Reasoning

The result of $CES(M, P, I, O_{r_j})$ in Equation 1 shows whether the model can *correctly predict the output* of $P$ for given inputs $I$. The results of $CES(M, P, I, V_{l_j})$, $CES(M, P, I, I_{l_j})$, $CES(M, P, I, B_{c_j})$, and $CES(M, P, I, P_{c_j})$ will be used to determine if the model's reasoning process is *valid* or *invalid*. The notion of reasoning validity enables CES to detect suspiciously correct output predictions under invalid reasoning process (§3.4), identify where the simulation diverges from ground truth (valid reasoning process followed by incorrect output prediction §3.4), (3) define a new self-consistency notion with respect to code reasoning (§4).

We determine the cases where code reasoning process is *invalid*, and consider the rest as valid. With respect to the properties that CES evaluates during execution simulation, we define three main cases of invalid reasoning process, evaluated using the equations below:

$$(CES(M, P, I, O_{r_j}) = 1) \quad \wedge \quad (\prod_{j=1}^{m+n} CES(M, P, I, X_{y_j}) = 0) \tag{2}$$

$$CES(M, P, I, P_{c_j}) \quad \neq \quad CES(M, P, I, B_{c_j}) \tag{3}$$

$$\exists \ j \ such \ that \ (CES(M, P, I, X_{y_j}) = 1) \wedge (\prod_{w=1}^{z} [\![M(P, I, X_{y_{j_w}}) = GT(P, I, X_{y_{j_w}})]\!] = 0) \tag{4}$$

At the highest level (represented by Equation 2), the reasoning is invalid if at least one of the intermediate predictions (Equation 1) is incorrect, but the output is correctly predicted. The illustrative example in Figure 2 shows such a case, which never happens in the real execution of the program, as the incorrect program state will be propagated to the output. As we will show (§5.4), such invalid reasoning process is common in LLMs due to hallucination, CoT shortcuts, and potential data leakage. There are other invalid reasoning process cases, regardless of the outcome of output prediction: for conditional statement, incorrect prediction of predicate and correct prediction of branch demonstrates invalid reasoning process (Equation 3); For compound properties, incorrect prediction of at least one sub-component and correct prediction of the compound also indicates invalid reasoning process (Equation 4). In the example of Figure 1, where GPT-4 Turbo mispredicts values of `odds`, if it correctly predicted the values of `zip(evens,odds)`, it could be considered an invalid reasoning.

## 3.4 Diagnosis

CES can promptly identify suspiciously correct outputs upon receiving the response from the model. To that end, it checks whether at least one of the conditions in Equations 2–4 exists in the response and marks the reasoning as invalid. For cases where the output prediction is correct for the simulation of $P$ under $I$, CES marks the correct output prediction as suspiciously correct for further investigation and excludes it from correct prediction results.

For the cases where the reasoning is *valid* but the output prediction is *incorrect*, CES identifies all the program points where a misprediction occurs ($CES(M, P, I, X_{y_j}) = 0$). Given that the reasoning process has been checked to be valid firsthand, the first incorrect prediction likely propagates in the execution simulation, causing the LLM to mispredict subsequent properties. Thereby, identifying

---

[2]The breakdown of the equation can be found in §A.5.

Figure 5: HumanEval/35 program (a), its corresponding control flow graph (b), and the prime path coverage of three test inputs for this program (c)

where the simulation execution starts to diverge from the real execution can reveal important information about the limitations of the models in code reasoning. CES follows a flow-sensitive static analysis on the program $P$, and selects the first statement with $CES(M, P, I, X_{y_j}) = 0$. Given that the reasoning is valid, the misprediction at this point will propagate, resulting in other intermediate incorrect properties prediction and, ultimately, incorrect output prediction. In the illustrative example of Figure 1, the misprediction of e results in subsequent incorrect output predictions.

## 4 THE SPECTRUM OF CODE EXECUTION REASONING

The code that LLMs generate should be comprehensively tested using tests that cover all the execution paths in the program. Otherwise, one can falsely claim an incorrect code to be correct. Similarly, one cannot claim victory on code reasoning until the LLM can correctly simulate all the execution paths. CES introduces the spectrum of code reasoning and evaluates the models based on their reasoning consistency across different execution paths.

Figure 5 shows a program with its control flow graph. Due to the existence of a loop, the number of execution paths for this program depends on the input. To account for this issue and having a bounded number of execution paths, CES considers *prime paths* in the control flow graph (Ammann & Offutt, 2017). A prime path in a cyclic graph is a path between two arbitrary nodes that does not visit any node more than once except for the starting and ending nodes. Figure 5-c shows three tests for the program of Figure 5-a and their prime path coverage during test execution. The union of these tests covers all the prime paths, hence, critical execution sequences in the program.

Given a program $P$, test inputs $I = \{i_1, \ldots, i_n\}$, and corresponding prime path coverage of tests $T_{cov} = \{cov_1, \ldots cov_n\}$ $(0 < cov_i \leq 1)$, we define the spectrum of code reasoning as follows:

**Definition 4. Strong Reasoning.** An LLM archives a strong level of code execution reasoning on program $P$, if it can consistently simulate the execution of $P$ under test inputs with *different* prime path coverage correctly.

$$\forall (p \neq q), j \quad (cov_p \neq cov_q) \wedge (CES(M, P, i_p, X_{y_j}) = CES(M, P, i_q, X_{y_j}) = 1) \quad (5)$$

**Definition 5. Weak Reasoning.** An LLM archives a weak level of code execution reasoning on program $P$, if it can only consistently simulate the execution of $P$ under test inputs with the *same* prime path coverage correctly.

$$(\forall (p \neq q), j \quad (cov_p = cov_q) \wedge (CES(M, P, i_p, X_{y_j}) = CES(M, P, i_q, X_{y_j}) = 1)) \quad \wedge$$
$$(\exists (r \neq p), k \quad (cov_r \neq cov_p) \wedge (CES(M, P, i_r, X_{y_k}) = 0)) \quad (6)$$

An LLM is randomly reasoning about program $P$, if there exists no consistency in correctly simulating the execution of test inputs regardless of their coverage.

For the example of Figure 5, GPT-4 Turbo consistently and correctly simulates the program execution across three tests, covering all the prime paths. As a result, it achieves a strong reasoning for this program. Our definition of strong reasoning does not require test inputs covering all the prime paths but only different ones. Even with such slack, we observe that even the frontier LLMs achieve strong reasoning for a handful of programs in the HumanEval benchmark (§5.3).

## 5 EVALUATION

To demonstrate the effectiveness of CES, we investigate the following research questions:

**RQ1: Performance in CES.** To what extent can LLMs simulate the program execution?

**RQ2: Reasoning Consistency Across Multiple Tests.** To what extent can LLMs consistently simulate the same or different execution path of the same program?

**RQ3: Incorrect and Suspiciously Correct Output Predictions.** At which program points are LLMs more likely to start diverging from the real execution? What are the potential root causes for incorrect and suspiciously correct output predictions?

**RQ4: Agreement Between CES and Other Programming Tasks.** Is there any correlation between the performance of LLMs in CES and programming tasks that inherently require control- and data-flow awareness?

### 5.1 EXPERIMENTAL SETUP

**Subject LLMs** We select thirteen pre-trained or instruction-tuned models of different sizes, covering both general-purpose and Code LLMs: GPT-4 Turbo (Achiam et al., 2023), Gemini-1.5 Pro (Team et al., 2023), CodeLlama (Roziere et al., 2023) (Base-7b, Instruct-7b, Base-13b, Instruct-13b, Instruc-34b), DeepSeekCoder (Bi et al., 2024) (Instruct-6.7b, Base-6.7b, Instruct-33b), Magicoder-S-6.7b (Wei et al., 2023), SemCoder-S (Ding et al., 2024b), and StarCoder2-15b (Lozhkov et al., 2024). We downloaded the open-access LLMs from HuggingFace (hug, 2024) and followed the best prompting practices from their official documents to ensure proper evaluation. Our experimental setting enforces temperature zero for all the models to ensure the reproducibility of results. For other parameters, we use the default setting of each model.

**Subject Programs** We evaluate subject LLMs on the HumanEval (Chen et al., 2021), the most widely used programming dataset of 164 Python programming problems. Our selection of this benchmark for evaluation is two-fold: First, evaluating the most recent Code LLMs under different programming tasks demonstrates a great performance on HumanEval. As we will show, such an outstanding performance is not necessarily due to code understanding, and many of them involve incorrect and unreasonable CoT shortcuts (§A.6); hence, it should not be considered a victory. Furthermore, HumanEval comes with extra artifacts, i.e., human-written bugs, which are required for RQ4 to evaluate the agreement between CES and bug-related tasks. The programs in HumanEval are mostly standalone methods but are challenging in terms of algorithmic complexity. Prior research also has shown that (Liu et al., 2024a) the mean and median cyclomatic complexity of the programs in this benchmark is close to other programming benchmarks. For each problem in the HumanEval dataset, we randomly selected three tests. When sampling, we did not control for coverage to be fair for all programs and include all of them in the evaluation.

### 5.2 RQ1. PERFORMANCE IN CES

Table 1 shows the detailed results of evaluating subject LLMs in code execution simulation. We break down the results into four categories of programs in the HumanEval dataset: programs with conditional statements only (CO), programs with loop statements only (LO), programs with both loops and conditional statements (LC), and programs with none of these programming constructs (Others). For the last category, the CES task becomes a simple output prediction; thereby, correct and incorrect intermediate reasoning is not applicable. When reporting the aggregated result (last four rows), we count them under *Valid Reasoning* rows, depending on the output prediction. We categorize our observations based on the validity of the reasoning as below:

- *Valid Reasoning Process.* LLMs are more likely to simulate the execution with valid (83.32%) than invalid reasoning process (16.68%). However, they mostly yield incorrect output predictions.

  **[Impact of size]**: Within the family of models, LLMs with more parameters always outperform smaller ones on correct output prediction: the performance improves from 18.40% (CodeLlama-Instruct-7b) to 25.17% (CodeLlama-Instruct-34b) and from 31.51% (DeepSeekCoder-Instruct-6.7b) to 39.43% (DeepSeekCoder-Instruct-33b). Between the models of different sizes, bigger models outperform smaller ones, except DeepSeekCoder-Instruct-34b slightly outperforming Gemini-1.5 Pro in correct output prediction.

  **[Impact of instruction-tuning]**: Instruction-tuning, although slightly, improves the performance of LLMs in code execution simulation: for CodeLlama-7b, CodeLlama-13b, and DeepSeekCoder-6.7b, the instruction-tuned version outperforms the base with the margin of 1.96%, 1.65%, and 3.47%, respectively. This is because the instruction-tuned versions follow prompt instructions better, which is important since code execution simulation is a complex task. For SemCoder (Ding et al., 2024b), fine-tuned on DeepSeekCoder-Base-6.7b with *execution data*, the improvement is

Table 1: Performance of LLMs in Code Execution Simulation (CES) task considering all the three sampled tests per each HumanEval program (total 164 programs). **CO**: programs with conditional statements, **LO**: programs with loops, **LC**: programs with both loops and conditional statements. We highlight the top three best-performing models with red ($1^{st}$), green ($2^{nd}$), and blue ($3^{rd}$).

| Programs | Predictions | | Subject LLMs | | | | | | | | | | | | |
|---|---|---|---|---|---|---|---|---|---|---|---|---|---|---|---|
| | | | CodeLlama | | | | | DeepSeekCoder | | | MagiCoder-S | SemCoder-S | StarCoder2 | Gemini-1.5 | GPT-4 |
| | | | (Inst-7b) | (Base-7b) | (Inst-13b) | (Base-13b) | (Inst-34b) | (Inst-6.7b) | (Base-6.7b) | (Inst-33b) | (6.7b) | (6.7b) | (15b) | (Pro) | (Turbo) |
| CO(24) | Valid | Correct Output | 23.37% | 20.90% | 25.97% | 23.38% | 30.88% | 38.89% | 37.50% | 41.67% | 29.17% | 47.22% | 40.28% | 54.17% | 81.94% |
| | Reasoning | Incorrect Output | 42.86% | 53.64% | 44.16% | 45.45% | 50.00% | 36.11% | 51.39% | 36.11% | 52.78% | 38.89% | 50.00% | 30.56% | 8.33% |
| | Invalid | Correct Output | 16.88% | 25.47% | 22.08% | 24.68% | 19.12% | 19.44% | 11.11% | 20.83% | 18.06% | 12.50% | 6.94% | 13.89% | 9.72% |
| | Reasoning | Incorrect Output | 16.88% | 0.00% | 7.79% | 6.49% | 0.00% | 0.00% | 0.00% | 1.39% | 0.00% | 1.39% | 2.78% | 1.39% | 0.00% |
| LO (12) | Valid | Correct Output | 8.11% | 0.00% | 40.54% | 35.14% | 35.29% | 47.22% | 38.89% | 52.78% | 41.67% | 41.67% | 40.54% | 38.89% | 55.56% |
| | Reasoning | Incorrect Output | 64.86% | 62.16% | 43.24% | 51.35% | 61.76% | 36.11% | 52.78% | 30.56% | 44.44% | 47.22% | 52.78% | 36.11% | 5.56% |
| | Invalid | Correct Output | 27.03% | 37.84% | 16.22% | 13.51% | 22.55% | 16.67% | 8.33% | 16.67% | 13.89% | 11.11% | 8.33% | 27.78% | 38.89% |
| | Reasoning | Incorrect Output | 0.00% | 0.00% | 0.00% | 0.00% | 0.00% | 0.00% | 0.00% | 0.00% | 0.00% | 0.00% | 0.00% | 0.00% | 0.00% |
| LC (75) | Valid | Correct Output | 3.69% | 2.46% | 7.37% | 4.15% | 6.86% | 11.56% | 7.11% | 15.56% | 11.56% | 10.22% | 16.00% | 12.44% | 36.00% |
| | Reasoning | Incorrect Output | 68.66% | 77.72% | 71.43% | 64.98% | 70.59% | 56.89% | 68.89% | 55.56% | 65.33% | 61.33% | 53.33% | 43.56% | 18.67% |
| | Invalid | Correct Output | 26.73% | 19.82% | 21.20% | 30.41% | 22.55% | 30.67% | 24.00% | 28.44% | 22.67% | 28.00% | 30.22% | 40.44% | 45.33% |
| | Reasoning | Incorrect Output | 0.88% | 0.00% | 0.00% | 0.44% | 0.00% | 0.89% | 0.00% | 0.44% | 0.44% | 0.44% | 0.44% | 1.78% | 0.00% |
| Others (53) | Correct Output | | 40.37% | 37.93% | 42.59% | 34.19% | 46.15% | 52.83% | 50.94% | 69.18% | 62.26% | 62.89% | 67.30% | 65.41% | 88.05% |
| | Incorrect Output | | 59.63% | 62.07% | 57.41% | 65.81% | 53.85% | 47.17% | 49.06% | 30.82% | 37.74% | 37.11% | 32.70% | 34.59% | 11.95% |
| Total (164) | Valid | Correct Output | 18.40% | 16.44% | 19.14% | 17.49% | 25.17% | 31.51% | 28.04% | 39.43% | 32.72% | 34.96% | 37.80% | 38.21% | 60.98% |
| | Reasoning | Incorrect Output | 61.85% | 66.69% | 61.11% | 67.07% | 61.47% | 49.28% | 58.75% | 42.88% | 53.05% | 49.39% | 46.14% | 38.22% | 14.02% |
| | Invalid | Correct Output | 16.67% | 16.87% | 18.52% | 14.20% | 13.37% | 18.09% | 13.21% | 17.28% | 14.02% | 15.45% | 15.45% | 22.56% | 25.00% |
| | Reasoning | Incorrect Output | 3.09% | 0.00% | 1.23% | 1.23% | 0.00% | 1.12% | 0.00% | 0.41% | 0.21% | 0.20% | 0.61% | 1.01% | 0.00% |

6.92%. SemCoder also outperforms instruction-tuned models of the same size or even bigger, demonstrating the impact of execution-aware fine-tuning in better code reasoning[3].

**[Impact of code]**: We can observe that LLMs struggle with loops (row *LO*) and complex programs with different code constructs (row *LC*), as the average correct output prediction for programs in these categories drop significantly compared to *CO* programs. Given the simplicity of programs in HumanEval compared to real-world code, this indicates LLMs may struggle more in reasoning about such complex code.

- *Invalid Reasoning Process.* Surprisingly, LLMs with good performance in valid reasoning process and correct output prediction also result in more invalid reasoning. Among all the models, GPT-4 Turbo and Gemini-1.5 Pro generate more invalid reasoning cases, 25% and 23.57%, respectively. In a similar trend, the base models that previously underperformed instruction-tuned models generate less suspiciously correct outputs. As we will discuss later with more details and in-depth analysis (§5.4), we speculate this is due to the interference of natural language reasoning and code reasoning. That is, instruction-tuned models that are better aligned with natural language instructions may *override* code reasoning with natural language reasoning shortcuts or hallucinations. The same observation holds for SemCoder, which generates more suspiciously correct predictions compared to DeepSeekCoder-Base-6.7b. While SemCoder incorporates execution information such as coverage, orders, and program states, it uses natural language monologues for instruction tuning. As a result, it falls into the same trap as other models and hallucinates with invalid reasoning process. We believe these results should initiate rethinking about pre-training or fine-tuning strategies for more realistic Code LLMs that can better reason about code.

## 5.3    RQ2. REASONING CONSISTENCY ACROSS MULTIPLE TESTS

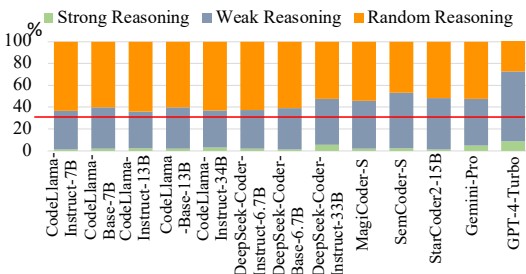

Figure 6: The reasoning strength of subject LLMs on HumanEval programs. The red line marks the percentage of subjects with at least two tests of different prime path coverage

To evaluate the extent to which LLMs can reason about different execution paths of the same program, we computed the percentage of programs they can *strongly*, *weakly*, or *randomly* simulate executions. We used Equations 5–6 and divided the values by the total number of programs in HumanEval. Figure 6 shows the result of this study. Given our random three-test selection process and since some programs are simple with no branches, only 52 out of 164 programs (31.71%) had at least two tests with different prime path coverage. The red line in Figure 6 marks this max number.

These results show that 55.59%, 41.69%, and 2.72% of reasoning, on average across all subject LLMs, are random, weak, and strong. The notion of self-consistency among different tests is simpler than self-consistency among different tasks (Min et al., 2023; Huang et al., 2023) since

---

[3]Per information in the paper, the authors have decontaminated HumanEval problems from their dataset.

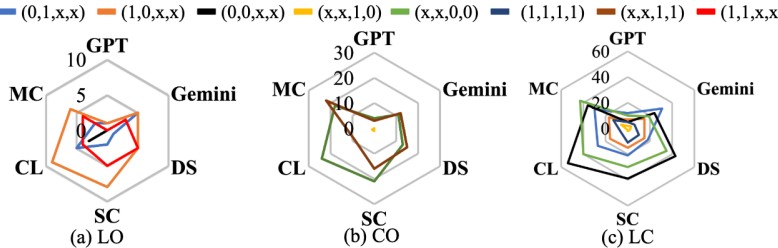

Figure 7: Comparison of simulation divergence locations for top six models across Loop Only (a), Condition Only (b), and Loops and Condition (c) programs. Models' names are abbreviated as follows: CL (CodeLlama-Instrcut-34b), DS (DeepSeek-Coder-Instruct-33b), Gemini (Gemini-1.5-Pro), GPT (GPT-4-Turbo), MC (MagiCoder-S), and SC (StarCoder2-15b). $\langle V_l, I_l, P_c, B_c \rangle$ labels represent location of divergence as $\langle$loop variable, loop iterator, conditional predicate, conditional branch$\rangle$. $x$ denotes the observation holds regardless of the values for a given variable

the task here is not changing. Yet, state-of-the-art LLMs cannot consistently simulate the same execution paths under different inputs. These results confirm that self-consistency across multiple tests is a vital evaluation metric and should be considered more seriously when evaluating LLM's performance in other programming tasks.

## 5.4 RQ3. INCORRECT AND SUSPICIOUSLY CORRECT OUTPUT PREDICTIONS

**Incorrect Output Prediction.** CES can automatically determine where the execution simulations diverge from the actual program execution. Figure 7 compares the simulation divergence locations across different categories of programs for the top six best-performing models from Table 1. We label the locations with quadruples $\langle V_l, I_l, P_c, B_c \rangle$, where the elements represent loop variable, loop iterator, conditional predicate, and conditional branch, respectively (§3.1). The values of 1 and 0 for an element indicate whether it is correctly predicted. The elements that are not applicable per program category or whose values do not impact the label are marked with an 'x.' The polygons inside the spider charts are mostly non-convex and overlapping, demonstrating different behaviors of the LLMs. In LO programs, mispredicting loop iterables—(1,0,x,x)—is the most common reason to initiate the divergence. For CO programs, mispredicting predicates and branches simultaneously—(x,x,0,0)—results in divergence. In LC programs, LLMs tend to mispredict both loop properties—(0,0,x,x), resulting in divergence from ground truth. We manually investigated these cases to understand *why* the predictions differed from ground-truth values. Our investigations reveal several shortcomings of the subject LLMs as follows: (1) LLMs fail to track loop iterable whose value is dynamically changing inside the loop (Listing 3 (HumanEval/13, DeepSeekCoder-Instruct-33b)); (2) LLMs may struggle to reason about compound properties, e.g., conditional statements with multiple sub-predicates (Listing 4 (HumanEval/57, Gemini-1.5 Pro)); (3) LLMs tend to hallucinate on branch decision making, i.e., although they correctly predict the predicate values, they mispredict if the branch will be taken or not (Listing 5 (HumanEval/148, GPT-4 Turbo)); (4) Nested constructs (if/loop inside another if/loop) can make it harder for LLMs to simulate program execution (Listing 6 (HumanEval/73, CodeLlama-Instruct-13b), Listing 7 (HumanEval/142, MagiCoder-S), and Listing 8(HumanEval/12, DeepSeekCoder-Instruct-6.7b)); (5) LLMs may struggle to reason about complex arithmetic/logic operations (Listing 9 (HumanEval/47, GPT-4 Turbo)); and (6) LLMs may mispredict return value of API calls (Listing 10 (HumanEval/160, Gemini-1.5 Pro)).

**Suspiciously Correct Output Prediction.** CES automatically detects suspiciously correct output predictions. However, it cannot explain *why* LLMs make such mistakes. We manually investigated them deeper to better understand the root causes of suspiciously correct output predictions. The most common culprit is the *CoT shortcut*. In such cases, a monologue-style step-by-step thinking of code execution seemingly overrides the code simulation asked in CES, making up for incorrect properties (loop variable, loop iterator, conditional predicate, branch) predictions and resulting in correct output prediction. The examples of such cases are shown in Figure 2 (HumanEval/156) and Listing 11 (HumanEval/11). In the latter, Gemini-1.5 Pro mispredicts the conditional predicate inside the loop ($i==j$) in iterations 3, 4, and 6. In the CoT reasoning, however, it looks at the method name (`string_xor`), assumes that the method implements bitwise `XOR`ing, and uses this assumption in the correct output prediction.

We also observed cases where simulation execution and CoT reasoning were incorrect, yet the LLM correctly predicted the output. Examples of such cases are Listing 12 (HumanEval/0), Listing 13

Table 2: The performance of LLMs on Bug Prediction, Bug Localization, Bug Repair, and CES. Cohen's Kappa coefficients represent the pairwise agreement between CES and these tasks.

| | CodeLlama-Instruct-34b | DeepSeek-Coder-Instruct -33b | MagiCoder-S | SemCoder-S | StarCoder2 -15b | Gemini-1.5-Pro | GPT-4-Turbo |
|---|---|---|---|---|---|---|---|
| **Bug Prediction** | 18.75% | 28.75% | 26.25% | 18.13% | 15.00% | 92.50% | 88.75% |
| **Bug Localization** | 41.88% | 40.63% | 24.38% | 44.38% | 33.13% | 72.50% | 71.25% |
| **Bug Repair** | 42.50% | 76.25% | 69.38% | 74.38% | 60.00% | 90.00% | 93.13% |
| **CES** | 18.13% | 27.50% | 20.61% | 23.75% | 25.00% | 32.50% | 55.00% |
| $\kappa_{CES,BP}$ | -0.06 | 0.05 | -0.02 | -0.06 | -0.07 | 0.09 | -0.02 |
| $\kappa_{CES,BL}$ | 0.02 | 0.18 | 0.03 | -0.02 | 0.05 | 0.06 | -0.02 |
| $\kappa_{CES,BR}$ | -0.04 | -0.03 | -0.01 | 0.02 | -0.03 | -0.02 | -0.01 |

(HumanEval/98), and Listing 14 (HumanEval/73). In the first example, GPT-4 Turbo mispredicts the inner loop variable values (idx1, elem2) as well as the conditional predicates (idx!=idx2 and distance < threshold). The CoT explains the code at a very high level without discussing the program states and variable values. Magically, in the end, it hallucinates the return value to be True. Similarly, in the second example, CodeLlama-Instruct-13b correctly predicts the output despite both simulation execution and CoT being incorrect. In the first two examples, the return type is boolean, making it easy for LLMs to predict output correctly with a high chance. The last example, however, returns an integer, which is harder to predict by chance. For this example, the CoT reasoning of GPT-4 Turbo shows an additional iteration over the loop. However, since this additional iteration does not change the output, the return value will be predicted correctly.

### 5.5 RQ4. AGREEMENT BETWEEN CES AND OTHER PROGRAMMING TASKS

LLMs should incorporate their knowledge of programming languages and examples they have seen to solve programming tasks. Otherwise, one cannot expect them to generalize to different tasks, perform reasonably on real-world programs, or trust them. We consider three programming tasks that, outside of the LLM world, performing them require execution awareness: bug prediction, bug localization, and bug repair. Ideally, if an LLM can correctly simulate an execution path, it is more likely to detect, localize, and repair a bug in that specific execution path. Similarly, we expect that if a model cannot correctly simulate an execution path, it fails to detect, localize, or fix the bug.

To investigate these two hypotheses, we used OctoPack (Muennighoff et al., 2023), a dataset of bugs generated by humans injected into HumanEval programs. We first executed OctoPack tests and identified the failed tests on the buggy version of HumanEval programs. We then checked if those were among our three sampled tests and kept the programs and tests passing the check. We prompted the seven best-performing models in CES with the prompts in §A.2 to perform bug prediction, localization, and repair.

Table 2 illustrates the performance of models in these tasks, as well as the Cohen's Kappa (McHugh, 2012) coefficients representing the pairwise agreement between CES and these tasks. Cohen's Kappa is a statistical test that checks the agreement between the two groups in terms of the validity of the property (how well Group A's validity predicts Group B's validity). It accounts for agreement by chance, making it more reliable than a simple percentage of agreement. Its coefficient ($\kappa$) takes a value between -1 and 1. The $\kappa = 1$ shows perfect agreement; $\kappa = 0$ indicates agreement by chance; negative values demonstrate no agreement or systematic disagreement.

From these results, we can see that there is, at best, a slight agreement between CES [4] and these tasks, rejecting our mentioned hypotheses. Figure 8 (§A.6 in Appendix) demonstrates the degree of overlap between CES and individual bug-related tasks across the programs. To better understand the reasons for agreements and disagreements, we manually investigated instances where models (1) succeeded in CES and other bug-related tasks and (2) failed in CES but succeeded in bug-related tasks, respectively. Our investigation shows that frontier models, e.g., GPT-4 Turbo and Gemini-1.5 Pro, attempt to simulate the execution when solving bug-related tasks. In cases of agreement with CES, their code execution simulation extracted from CoT is correct (Figure 9). In case of disagreement, natural language hallucinations interfere with their code execution simulation (Figure 10), or their code execution simulation is incorrect, and they succeed in the task through shortcuts (Figure 11). **The results of current and prior research questions should bring the attention to the following questions: Shall we settle on LLMs that incorporate pattern matching or similar strategies but not inductive code reasoning when performing programming tasks? To what extent can we trust such LLMs?**

---

[4] The CES values are different from that of in Table 1, since this experiment was done on buggy programs in the OctoPack dataset.

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

# A APPENDIX

## A.1 PROMPT DESIGN OF CES

Figure 14 shows the detailed prompt template used in CES, consisting of five key components: the system message, adaptive in-context learning examples, instruction, question, and response. In both the question and the in-context learning examples, we begin by commenting on the program using "sub-questions," which include variables or expressions related to loop or conditional construct properties, along with placeholders marked as "??". Within the in-context learning examples, we also provide sample answers where the placeholders are replaced with the correct values or states. The instruction specifies how the LLMs should respond to the questions: answers related to loop predictions should be enclosed in "[STATE]" and "[/STATE]", condition-related predictions in "[CONDITION]" and "[/CONDITION]", and branch predictions in "[BRANCH]" and "[/BRANCH]". In the question section, we annotate the program similarly to how it is done in the in-context learning examples, and we expect the LLMs to generate responses in the format outlined in the instruction.

With the growing capabilities of large language models (LLMs), in-context learning (Dong et al., 2022) has emerged as an important paradigm for natural language processing (NLP) tasks. Moreover, research (Wu et al., 2022; Wan et al., 2023) suggests that selecting well-performing in-context learning examples tailored to different inputs can enhance LLMs' ability to generate correct outputs. Building on the concept of adaptively selecting in-context learning examples, we developed **11** sets of examples based on program constructs and their locations: **if**, **elif**, **nested if**, **for loop**, **while loop**, **nested loop**, **if inside while loop**, **if outside while loop**, **if inside for loop**, **if outside for loop**, and **if inside nested loop**. For each problem in the benchmark, we utilize the tree-sitter [5] to extract code constructs and their locations, and then select the corresponding examples from our predesigned in-context learning candidates.

Table 3: Ablation Study on Prompting Strategies.

|  |  | GPT-4-Turbo | | | Gemini-1.5-Pro | | | DeepSeekCoder-Inst-33b | | | StarCoder2-15b | | | SemCoder-S | | |
|---|---|---|---|---|---|---|---|---|---|---|---|---|---|---|---|---|
|  |  | CES | CES (-CoT) | CES (Fixed ICL) | CES | CES (-CoT) | CES (Fixed ICL) | CES | CES (-CoT) | CES (Fixed ICL) | CES | CES (-CoT) | CES (Fixed ICL) | CES | CES (-CoT) | CES (Fixed ICL) |
| **Valid Reasoning** | **Correct Output** | 60.98% | 55.49% | 57.93% | 38.21% | 31.71% | 29.88% | 39.43% | 36.18% | 33.54% | 37.80% | 34.96% | 35.16% | 34.96% | 32.52% | 30.49% |
|  | **Incorrect Output** | 14.02% | 20.73% | 17.07% | 38.22% | 48.17% | 46.95% | 42.88% | 46.34% | 49.39% | 46.14% | 50.41% | 49.19% | 49.39% | 51.22% | 53.86% |
| **Invalid Reasoning** | **Correct Output** | 25.00% | 21.95% | 22.56% | 22.56% | 18.29% | 19.51% | 17.28% | 16.26% | 15.24% | 15.45% | 14.02% | 14.43% | 15.45% | 14.84% | 13.82% |
|  | **Incorrect Output** | 0.00% | 1.83% | 2.44% | 1.01% | 1.83% | 3.66% | 0.41% | 1.22% | 1.83% | 0.61% | 0.61% | 1.22% | 0.20% | 1.42% | 1.83% |

To further demonstrate the effectiveness of our design choice, we repeated the experiments for the top five best-performing models with two variants of CES:

---

[5] https://tree-sitter.github.io/tree-sitter/creating-parsers

- **CES (-CoT):** Here we remove the natural language CoT in our adaptive in-context learning (ICL) examples (text between [REASONING] and [/REASONING] in Figure 14) and ask LLMs to include only the annotated code and the output of code execution in the response.
- **CES (Fixed ICL)**: instead of selecting ICL examples adaptively according to the program constructs and their locations, we use the fixed ICL example (shown in Listing 1) for all the programs in the benchmark.

Table 3 shows the results of this experiment. Our most notable observations are:

- Comparing **CES** with **CES(-CoT)**, we can see that including CoT can improve valid reasoning process & correct output prediction by 4.11%. Comparing **CES** with **CES (Fixed ICL)**, we find that providing adaptive ICL examples instead of fixed ones can improve correct code reasoning by 4.88% on average.
- Including the CoT and applying adaptive prompting also result in more invalid reasoning cases. In particular, they improve the invalid reasoning process & correct output by 2.32% and 2.04%, respectively. However, the gain in invalid reasoning process & correct output is still smaller than that of invalid reasoning process & correct output, which still supports our prompt design.

These observations support our prior claim about the inference of natural language reasoning, which is prone to shortcuts and hallucinations, with more formal reasoning in CES. We believe that, once the models replace their natural language CoT reasoning with more formal ones, including the formal CoT reasoning results in benefits without any side effects.

In Listing 1 and Listing 2, we show how adaptive in-context learning example improves DeepSeekCoder-Inst-33b on HumanEval/132. HumanEval/132 (is_nested) has if-else construct inside the for loop, and in fixed-ICL setting (Listing 1), we present a program with for loop (ICL Example 1) and another program with if statement (ICL Example 2) in the prompt and expect them to teach the model to respond correctly. However, we can see that DeepSeekCoder-Inst-33b still fails to track the states of conditional predicates inside the for loops and makes incorrect predictions on the loop properties of the second for loop. In Listing 2, according to the program construct of HumanEval/132, we include an ICL example that also has a conditional statement inside the for loop in the prompt, and it helps DeepSeekCoder-Inst-33b to generate the correct reasoning process and correct output.

Listing 1: Fixed ICL Example (HumanEval/132)

```
You are given a piece of Python code and its output. You are asked to complete the
following task:
1. Replace the "??" in the comment with the state of a variable or the return value of a
method call/expression. Enclose the annotated code with [ANSWER] AND [/ANSWER]
2. Think step by step and print your reasoning process of the code execution.
3. Print the output and enclose it with [OUTPUT] and [/OUTPUT].

## ICL Example 1
[CODE]
def sum_of_integer(N):
    sum_1 = 0
    for i in range(1,N+1): ## [STATE]i=??[/STATE][STATE]range(1,N+1)=??[/STATE][STATE]N+1=??[/
        STATE]
        sum += i
    return sum
[/CODE]
[INPUT]
sum_of_integer(5)
[/INPUT]

[REASONING]
The function sum_of_integer is called with ( N = 5 ).
Inside the function, a variable `sum` is initialized to 0.
The loop iterates from 1 through 5 (since ( N = 5 )).
First Iteration (i = 1):
    `sum += i` updates `sum` to 1 ( 0 + 1 = 1 ).
Second Iteration (i = 2):
    `sum += i` updates `sum` to 3 ( 1 + 2 = 3 ).
Third Iteration (i = 3)
    `sum += i` updates `sum` to 6 ( 3 + 3 = 6 ).
Fourth Iteration (i = 4)
    `sum += i` updates `sum` to 10 ( 6 + 4 = 10 ).
Fifth Iteration (i = 5)
```

```
    `sum += i` updates `sum` to 15 ( 10 + 5 = 1 ).
After the loop completes, the function returns the value of `sum`, which is now 15.
[/REASONING]

[ANSWER]
def sum_of_integer(N):
    sum_1 = 0
    for i in range(1,N+1): ## [STATE]i=[1, 2, 3, 4, 5][/STATE][STATE]range(1,N+1)=[1, 2, 3, 4,
        5][/STATE][STATE]N+1=[6][/STATE]
        sum += i
    return sum
[/ANSWER]

[OUTPUT]
15
[/OUTPUT]

## ICL Example 2
[CODE]
def check_integer(i):
    if i >= 1 and i <3: ## [CONDITION](i>=1) = ??[/CONDITION][CONDITION](i <3) = ??[/CONDITION
        ][CONDITION](i >= 1 and i <3)=??[/CONDITION][BRANCH]taken=??[/BRANCH]
        return 0
    elif i < 5: ## [CONDITION] (i<5) = ??[/CONDITION][BRANCH]taken=??[/BRANCH]
        return 1
    else: ## [BRANCH]taken=??[/BRANCH]
        return 2
[/CODE]

[INPUT]
check_integer(4)
[/INPUT]

[REASONING]
The function check_integer is called with ( i = 4 ).
i is 2, which doesn't satisfy ( 1 <= i < 3 ).The condition i >= 1 and i < 3 is False.
The first branch is not taken.
i = 2 satisfies (i<2). The condition i<5 is True, and the second branch is taken.
Therefore, the final output of the function check_integer(2) is 1.
[/REASONING]

[ANSWER]
def check_integer(i):
    if i >= 1 and i <3: ## [CONDITION](i>=1) = [True][/CONDITION][CONDITION](i <3) = [True][/
        CONDITION][CONDITION](i >= 1 and i <3)=[True][/CONDITION][BRANCH]taken=[Y][/BRANCH]
        return 0
    elif i < 5:
        return 1
    else: ## [BRANCH]taken=[N][/BRANCH]
        return 2
[/ANSWER]

[OUTPUT]
1
[/OUTPUT]

## Question
[CODE]
def is_nested(string):
    opening_bracket_index = []
    closing_bracket_index = []
    for i in range(len(string)): ## [STATE]i=??[/STATE][STATE]range(len(string))=??[/STATE][
        STATE]len(string)=??[/STATE]
        if string[i] == '[': ## [CONDITION](string[i] == '[')=??[/CONDITION][BRANCH]taken=??[/
            BRANCH]
            opening_bracket_index.append(i)
        else: ## [BRANCH]taken=??[/BRANCH]
            closing_bracket_index.append(i)
    closing_bracket_index.reverse()
    cnt = 0
    i = 0
    l = len(closing_bracket_index)
    for idx in opening_bracket_index: ## [STATE]idx=??[/STATE][STATE]opening_bracket_index
        =??[/STATE]
        if i < l and idx < closing_bracket_index[i]: ## [CONDITION](i < l and idx <
            closing_bracket_index[i])=??[/CONDITION][CONDITION](i < l)=??[/CONDITION][
            CONDITION](idx < closing_bracket_index[i])=??[/CONDITION][BRANCH]taken=??[/BRANCH
            ]
            cnt += 1
            i += 1
    return cnt >= 2
```

```
[/CODE]

[INPUT]
is_nested('[[]]')
[/INPUT]
### Response
[REASONING]
1. The function is_nested(string) is defined. It takes one parameter, string.
2. Inside the function, two empty lists are initialized: opening_bracket_index and
closing_bracket_index. These lists will be used to store the indices of opening and closing
brackets in the string.
3. The for loop runs from 0 to the length of the string.
Loop details when string='[[]]':
When string[i]='[': opening_bracket_index becomes [0].
When string[i]=']': closing_bracket_index becomes [2].
4. The closing_bracket_index list is reversed.
5. Two variables cnt and i are initialized to 0. cnt is used to count the number of nested
brackets, and i is used to iterate over the closing_bracket_index list.
6. The for loop iterates over the opening_bracket_index list. For each index idx in
opening_bracket_index, it checks if idx is less than the current index i in
closing_bracket_index. If it is, it increments cnt and i.
7. Finally, the function returns True if cnt is greater than or equal to 2, and False
otherwise. This indicates whether the string contains at least two nested brackets.
[/REASONING]

[ANSWER]
def is_nested(string):
    opening_bracket_index = []
    closing_bracket_index = []
    for i in range(len(string)): ## [STATE]i=[0, 1, 2, 3][/STATE][STATE]range(len(string))=[0,
        1, 2, 3][/STATE][STATE]len(string)=[4][/STATE]
        if string[i] == '[':
            ## [CONDITION](string[i] == '[')=[False][/CONDITION][BRANCH]taken=[N][/BRANCH]
            opening_bracket_index.append(i)
        else: ## [BRANCH]taken=[Y][/BRANCH]
            closing_bracket_index.append(i)
    closing_bracket_index.reverse()
    cnt = 0
    i = 0
    l = len(closing_bracket_index)
    for idx in opening_bracket_index:
        ## [STATE]idx=[0][/STATE][STATE]opening_bracket_index=[0][/STATE]
        if i < l and idx < closing_bracket_index[i]:
            ## [CONDITION](i < l and idx < closing_bracket_index[i])=[True][/CONDITION]
        [CONDITION](i < l)=[True][/CONDITION]
        [CONDITION](idx < closing_bracket_index[i])=[True][/CONDITION]
        [BRANCH]taken=[Y][/BRANCH]
            cnt += 1
            i += 1
    return cnt >= 2
[/ANSWER]

[OUTPUT]
False
[/OUTPUT]
```

Listing 2: Adapative ICL Example (HumanEval/132)

```
You are given a piece of Python code and its output. You are asked to complete the following
task:
1. Replace the "??" in the comment with the state of a variable or the return value of a
method call/expression. Enclose the annotated code with [ANSWER] AND [/ANSWER]
2. Think step by step and print your reasoning process of the code execution.
3. Print the output and enclose it with [OUTPUT] and [/OUTPUT].

## Adaptive ICL Example
[CODE]
def sum_of_integer(N):
    sum_1 = 0
    for i in range(1,N+1): ## [STATE]i=??[/STATE][STATE]range(1,N+1)=??[/STATE][STATE]N+1=??[/
        STATE]
        if i >= 1 and i <3: ## [CONDITION](i>=1) = ??[/CONDITION][CONDITION](i <3) = ??[/
            CONDITION][CONDITION](i >= 1 and i <3)=??[/CONDITION][BRANCH]taken=??[/BRANCH]
            sum_1 += 1
        elif i < 5: ## [CONDITION](i<5)=??[/CONDITION][BRANCH]taken=??[/BRANCH]
            sum_1+= 2
        else: ## [BRANCH]taken=??[/BRANCH]
                sum_1 *= 4
    return sum_1
[/CODE]
```

```
[INPUT]
sum_of_integer(5)
[/INPUT]

[REASONING]
The function sum_of_integer is called with ( N = 5 ).
Inside the function, a variable sum_1 is initialized to 0.
The loop iterates from 1 through 5 (since ( N = 5 )).
First Iteration (i = 1)
    - i is 1, which satisfies ( 1 <= i < 3 ).
    - The condition i >= 1 and i < 3 is true.
    - Action: sum_1 += 1 results in sum_1 = 1.
Second Iteration (i = 2)
    - i is 2, which still satisfies ( 1 <=i < 3 ).
    - The same condition is true.
    - Action: sum_1 += 1 updates sum_1 to 2 (1 + 1 = 2).
Third Iteration (i = 3)
    - i is 3, which now satisfies ( 3 <= i < 5 ).
    - The condition i < 5 is true (while the first condition i >= 1 and i < 3 is false).
    - Action: sum_1 += 2 updates sum_1 to 4 (2 + 2 = 4).
Fourth Iteration (i = 4)
    - i is 4, also satisfying ( 3 <= i < 5 ).
    - The condition i < 5 is still true.
    - Action: sum_1 += 2 updates sum_1 to 6 (4 + 2 = 6).
Fifth Iteration (i = 5)
    - i is 5, now outside the previous conditions, falling into the else block.
    - The else block's action: sum_1 *= 4 updates sum_1 to 24 (6 * 4 = 24).
Return Statement
    - After the loop completes, the function returns the value of sum_1, which is now 24.

Therefore, the final output of the function sum_of_integer(5) is 24.
[/REASONING]

[ANSWER]
def sum_of_integer(N):
    sum_1 = 0
    for i in range(1,N+1): ## [STATE]i=[1, 2, 3, 4, 5][/STATE][STATE]range(1,N+1)=[1, 2, 3, 4,
         5][/STATE][STATE]N+1=[6][/STATE]
        if i >= 1 and i <3: ## [CONDITION](i>=1) = [True, True, True, True, True][/CONDITION][
             CONDITION](i <3 )= [True, True, False, False, False][/CONDITION][CONDITION](i >=
             1 and i <3)=[True, True, False, False, False][/CONDITION][BRANCH]taken=[Y, Y, N,
             N, N][/BRANCH]
            sum_1 += 1
        elif i < 5: ## [CONDITION](i<5)=[True, True, True, True, False][/CONDITION][BRANCH]
             taken=[N, N, Y, Y, N][/BRANCH]
            sum_1+= 2
        else: ## [BRANCH]taken=[N, N, N, N, Y][/BRANCH]
                sum_1 *= 4
    return sum_1
[/ANSWER]

[OUTPUT]
24
[/OUTPUT]

## Question
[CODE]
def is_nested(string):
    opening_bracket_index = []
    closing_bracket_index = []
    for i in range(len(string)): ## [STATE]i=??[/STATE][STATE]range(len(string))=??[/STATE][
         STATE]len(string)=??[/STATE]
        if string[i] == '[': ## [CONDITION](string[i] == '[')=??[/CONDITION][BRANCH]taken=??[/
             BRANCH]
            opening_bracket_index.append(i)
        else: ## [BRANCH]taken=??[/BRANCH]
            closing_bracket_index.append(i)
    closing_bracket_index.reverse()
    cnt = 0
    i = 0
    l = len(closing_bracket_index)
    for idx in opening_bracket_index: ## [STATE]idx=??[/STATE][STATE]opening_bracket_index
         =??[/STATE]
        if i < l and idx < closing_bracket_index[i]: ## [CONDITION](i < l and idx <
             closing_bracket_index[i])=??[/CONDITION][CONDITION](i < l)=??[/CONDITION][
             CONDITION](idx < closing_bracket_index[i])=??[/CONDITION][BRANCH]taken=??[/BRANCH
             ]
            cnt += 1
            i += 1
```

```
    return cnt >= 2
[/CODE]

[INPUT]
is_nested('[[]]')
[/INPUT]
### Response
[REASONING]
The function is_nested is called with ( string = '[[]]' ).
Inside the function, two empty lists are initialized: opening_bracket_index and
closing_bracket_index.

Iteration 1:
    -The character '[' is found at index 0, so it is appended to opening_bracket_index.
Iteration 2:
    -The character '[' is found at index 1, so it is appended to opening_bracket_index.
Iteration 3:
    -The character ']' is found at index 2, so it is appended to closing_bracket_index.
Iteration 4:
    -The character ']' is found at index 3, so it is appended to closing_bracket_index.

After the loop, the closing_bracket_index list is reversed.

Iteration 1:
    -The index 3 is less than the length of closing_bracket_index (4) and is less than the
    element at index 0 of closing_bracket_index (2).
    -The condition is true, so cnt is incremented by 1 and i is incremented by 1.
Iteration 2:
    -The index 2 is less than the length of closing_bracket_index (4) and is less than the
    element at index 1 of closing_bracket_index (3).
    -The condition is true, so cnt is incremented by 1 and i is incremented by 1.

After the loop, cnt is 2, which is greater than or equal to 2, so the function returns True.
[/REASONING]

[ANSWER]
def is_nested(string):
    opening_bracket_index = []
    closing_bracket_index = []
    for i in range(len(string)): ## [STATE]i=[0, 1, 2, 3][/STATE][STATE]range(len(string))
        =[4][/STATE][STATE]len(string)=[4][/STATE]
        if string[i] == '[': ## [CONDITION](string[i] == '[')=[True, True, False, False][/
            CONDITION][BRANCH]taken=[Y, Y, N, N][/BRANCH]
            opening_bracket_index.append(i)
        else: ## [BRANCH]taken=[N, N, Y, Y][/BRANCH]
            closing_bracket_index.append(i)
    closing_bracket_index.reverse()
    cnt = 0
    i = 0
    l = len(closing_bracket_index)
    for idx in opening_bracket_index: ## [STATE]idx=[0, 1][/STATE][STATE]opening_bracket_index
        =[0, 1][/STATE]
        if i < l and idx < closing_bracket_index[i]: ## [CONDITION](i < l and idx <
            closing_bracket_index[i])=[True, True][/CONDITION][CONDITION](i < l)=[True, True
            ][/CONDITION][CONDITION](idx < closing_bracket_index[i])=[True, True][/CONDITION
            ][BRANCH]taken=[Y, Y][/BRANCH]
            cnt += 1
            i += 1
    return cnt >= 2

[/ANSWER]

[OUTPUT]
True
[/OUTPUT]
```

## A.2 PROMPT TEMPLATE FOR BUG PREDICTION, LOCALIZATION, AND REPAIR

### A.2.1 PROMPT TEMPLATE FOR BUG PREDICTION

```
Given the following code, tests, and natural language specification, is the code buggy or not?
First, think step by step and simulate the execution process of the test.
Then respond with "Yes" or "No". "Yes" menas the code is buggy, "No" means it is not buggy.
Enclose your response with "```response" and "```".
Code:
from typing import List
def below_zero(operations: List[int]) -> bool:
    balance = 0
```

```
    for op in operations:
        balance += op
        if balance == 0:
            return True
    return False
```

**Test:**
```
assert below_zero([1, -2, 2, -2, 5, -5, 4, -4]) == True
```

**The code should satisfy the following natural language specification:**
Write a Python function below_zero(operations: List[int]) -> bool to solve the following
problem: You are given a list of deposit and withdrawal operations on a bank account that
starts with zero balance. Your task is to detect if at any point the balance of account
fallls below zero, and at that point function should return True. Otherwise it should
return False.

**Reasoning**
[Reasoning]
Now let's look at the program. The function initializes `balance` to 0. It iterates through
each operation in the operations list, adjusting balance by adding each operation (op).
It checks if balance == 0 (which only detects when the balance exactly reaches zero, not when
it goes below zero). If balance == 0, the function returns True, though we're actually
interested in whether balance < 0.
In the test case `operation` is initialized with [1, -2, 2, -2, 5, -5, 4, -4].
In the first iteration, op=1, balance=1.
In the second iteration, op=-2, balance=-1.
In the third iteration, op=2, balance=1.
In the forth iteration, op=-2, balance=-1.
In the fifth iteration, op=5, balance=4.
In the sixth iteration, op=-5, balance=-1.
In the seventh iteration, op=4, balance=3.
In the eighth iteration, op=-4, balance=-1.
`balance` never exactly reaches zero at a point when it first goes negative and the function
will return False as the output while the expected output is True.
Therefore the code is buggy.
[/Reasoning]
**Response:**
'''response
Yes
'''

**Code:**
{code}
**Test:**
{test}
**The code should satisfy the following natural language specification:**
{nl}

### A.2.2 PROMPT TEMPLATE FOR BUG LOCALIZATION

```
Given the following buggy code, test, and natural language specification, your task is to
identify the buggy line.
First, think step by step and simulate the execution of the provided test.
Then print the buggy line. Enclose your response with "'''response" and "'''".
```
**Buggy Code:**
```
from typing import List
def below_zero(operations: List[int]) -> bool:
    balance = 0
    for op in operations:
        balance += op
        if balance == 0:
            return True
    return False
```

**Test:**
```
assert below_zero([1, -2, 2, -2, 5, -5, 4, -4]) == True
```

**The code should satisfy the following natural language specification:**
Write a Python function below_zero(operations: List[int]) -> bool to solve the following
problem: You are given a list of deposit and withdrawal operations on a bank account that
starts with zero balance. Your task is to detect if at any point the balance of account
fallls below zero, and at that point function should return True. Otherwise it should
return False.

**Reasoning:**
[Reasoning]
Now let's look at the program. The function initializes `balance` to 0. It iterates through
each operation in the operations list, adjusting balance by adding each operation (op).
It checks if balance == 0 (which only detects when the balance exactly reaches zero, not when

```
it goes below zero). If balance == 0, the function returns True, though we're actually
interested in whether balance < 0.
In the test case 'operation' is initialized with [1, -2, 2, -2, 5, -5, 4, -4].
In the first iteration, op=1, balance=1.
In the second iteration, op=-2, balance=-1.
In the third iteration, op=2, balance=1.
In the forth iteration, op=-2, balance=-1.
In the fifth iteration, op=5, balance=4.
In the sixth iteration, op=-5, balance=-1.
In the seventh iteration, op=4, balance=3.
In the eighth iteration, op=-4, balance=-1.
'balance' never exactly reaches zero at a point when it first goes negative and the function
will return False as the output while the expected output is True.
Therefore the buggy line is ' if balance == 0:', which can not catch any negative balance.
[/Reasoning]

Response:
'''response
        if balance == 0:
'''

Buggy Code:
{buggy code}
Test:
{test}
The code should satisfy the following natural language specification:
{nl}
```

### A.2.3  PROMPT TEMPLATE FOR BUG REPAIR

```
Fix bugs in below_zero.
First, think step by step and simulate the execution of the provided test.
Enclose the fixed code in '''python and '''.

Buggy Code:
def below_zero(operations: List[int]) -> bool:
    balance = 0
    for op in operations:
        balance += op
        if balance == 0:
            return True
    return False

Test:
assert below_zero([1, -2, 2, -2, 5, -5, 4, -4]) == True

The fixed code should satisfy the following natural language specification:
Write a Python function below_zero(operations: List[int]) -> bool to solve the following
problem: You are given a list of deposit and withdrawal operations on a bank account that
starts with zero balance. Your task is to detect if at any point the balance of account
fallls below zero, and at that point function should return True. Otherwise it should
return False.

Reasoning
[Reasoning]
Now let's look at the program. The function initializes 'balance' to 0. It iterates through
each operation in the operations list, adjusting balance by adding each operation (op).
It checks if balance == 0 (which only detects when the balance exactly reaches zero, not when
it goes below zero). If balance == 0, the function returns True, though we're actually
interested in whether balance < 0.
In the test case 'operation' is initialized with [1, -2, 2, -2, 5, -5, 4, -4].
In the first iteration, op=1, balance=1.
In the second iteration, op=-2, balance=-1.
In the third iteration, op=2, balance=1.
In the forth iteration, op=-2, balance=-1.
In the fifth iteration, op=5, balance=4.
In the sixth iteration, op=-5, balance=-1.
In the seventh iteration, op=4, balance=3.
In the eighth iteration, op=-4, balance=-1.
'balance' never exactly reaches zero at a point when it first goes negative and the function
will return False as the output while the expected output is True.
However, if we were correctly checking 'balance < 0', the function would catch any negative
balance.
[/Reasoning]

Fixed Code:
'''python
def below_zero(operations: List[int]) -> bool:
    balance = 0
```

```
    for op in operations:
        balance += op
        if balance < 0:
            return True
    return False
'''

Fix bugs in {method_name}.
First, think step by step and simulate the execution of the provided test.
Enclose the fixed code in '''python and '''.

Buggy Code:
{buggy code}
Test:
{test}
The fixed code should satisfy the following natural language specification:
{nl}
```

## A.3 CASE STUDY

This section lists the results of our manual investigation into the root causes of incorrect output predictions and suspiciously correct output predictions. Each listing corresponds to a program in the HumanEval dataset and a specific model that attempted to simulate its execution given the specified inputs. We highlight the mispredictions in red and the correct predictions in green. Each listing also includes the ground-truth values for program properties that are mispredicted.

Listing 3: HumanEval/13

```
def greatest_common_divisor(a: int, b: int) -> int:
    while b: ## [STATE]b=[60, 144, 0][/STATE]
        a, b = b, a % b
    return a
greatest_common_divisor(144, 60)
```

**Ground Truth:**
b=[60, 24, 12, 0], output=12

**DeepSeek-Coder-Instruct-33b**
Predicted output: 60
**Analysis:**
First Iteration:
- b != 0, so the loop continues; _a = 60, b = 144 % 60 = 144._
- a and b are updated to b and a % b respectively. Now, a = 144, b = 60.
Second Iteration:
- b != 0, so the loop continues. a = 60, b = 144 % 60 = 0.
- a and b are updated to b and a % b respectively. Now, a = 0, b = 60.
The loop terminates because b is 0.
The function returns the value of "a", which is now **60**.

Listing 4: HumanEval/57

```
def monotonic(l: list):
    if l == sorted(l) or l == sorted(l, reverse=True):
##[CONDITION](l == sorted(l))=False[/CONDITION]
##[CONDITION](l == sorted(l,reverse=True))=False[/CONDITION]
##[CONDITION](l == sorted(l) or l == sorted(l,reverse=True))=False[/CONDITION]
## [BRANCH]taken=[N][/BRANCH]
        return True
    return False
monotonic([4, 1, 0, -10])
```

**Ground Truth:**
(l ==  sorted(l)) = False
(l == sorted(l, reverse=True)) = True
(l == sorted(l) or l == sorted(l, reverse=True)) = True
output: True

**Gemini-1.5-Pro**
Predicted output: False
**Analysis:**
The function 'monotonic' is called with the list "l = [4, 1, 0, -10]".
The function checks if the list is sorted in ascending order or in descending order.
The condition 'l == sorted(l)' checks if the list is sorted in ascending order.
The condition 'l == sorted(l, reverse=True)' checks if the list is sorted in descending order.
The condition 'l == sorted(l or l == sorted(l, reverse=True)' checks if the list is sorted in
     either ascending or descending order.

The condition is false because the list is not sorted in either ascending or descending order.
Therefore, the function returns False.

### Listing 5: HumanEval/148

```
def bf(planet1, planet2):
    planet_names = ("Mercury", "Venus", "Earth", "Mars", "Jupiter", "Saturn", "Uranus", "
        Neptune")
    if planet1 not in planet_names or planet2 not in planet_names or planet1 == planet2: ## [
        CONDITION](planet1 not in planet_names or planet2 not in planet_names or planet1 ==
        planet2)=[False][/CONDITION][CONDITION](planet1 == planet2)=[False][/CONDITION][
        CONDITION](planet1 not in planet_names)=[False][/CONDITION][CONDITION](planet2 not in
         planet_names)=[False][/CONDITION]
[BRANCH]taken=[Y][/BRANCH]
        return ()
    planet1_index = planet_names.index(planet1)
    planet2_index = planet_names.index(planet2)
    if planet1_index < planet2_index: ## [CONDITION](planet1_index < planet2_index)=[True][/
        CONDITION]
[BRANCH]taken=[N][/BRANCH]
        return (planet_names[planet1_index + 1: planet2_index])
    else: ## [BRANCH]taken=[N][/BRANCH]
        return (planet_names[planet2_index + 1 : planet1_index])
bf("Jupiter", "Neptune")
```

**Ground Truth:**
first if branch: [N]; second if branch: [Y]
output: ('Saturn', 'Uranus')

**GPT-4-Turbo:**
predicted output: ()

**Analysis:**
The function bf is called with ( planet1 = "Jupiter", planet2 = "Neptune" ).
The first condition 'planet1 not in planet_names or planet2 not in planet_names or planet1 ==
planet2' is false, since all the planets are present in the list. The second condition
'planet1 == planet2' is also false, since planet1 and planet2 are different. The third
condition planet1 not in planet_names is false, since planet1 is present in the list.
The fourth condition planet2 not in planet_names is also false, since planet2 is present in
the list. Therefore, the if branch is taken and the method returns (). Therefore, the final
output of the function `bf("Jupiter", "Neptune")` is ().

### Listing 6: HumanEval/73

```
def smallest_change(arr):
    ans = 0
    for i in range(len(arr) // 2): ## [STATE]i=[0, 1, 2, 3][/STATE]
## [STATE]range((len(arr) // 2))=[0, 1, 2, 3][/STATE][STATE](len(arr) // 2)=4[/STATE]
        if arr[i] != arr[len(arr) - i - 1]:
## [CONDITION](arr[i] != arr[len(arr) - i - 1])=[True, True, False, False][/CONDITION]
##[BRANCH]taken=[Y, Y, Y, N][/BRANCH]
            ans += 1
    return ans

smallest_change([1, 2, 3, 4, 3, 2, 2])
```
**Ground Truth:**
i = [0,1,2]
range((len(arr) // 2)) = [0,1,2]
(len(arr) // 2) = 3
output: 1

**CodeLlama-Instruct-13b:**
Predicted output: 2
**Analysis:**
Iteration 1:
    - `i` is 0, `arr[i]` is 1, and arr[len(arr) - i - 1] is 2.
    - The condition `arr[i] != arr[len(arr) - i - 1]` is true, so `ans` becomes 1.
Iteration 2:
    - i is 1, `arr[i]` is 2, and `arr[len(arr) - i - 1]` is 3. ......
Iteration 3:
    - i is 2, arr[i] is 3, and `arr[len(arr) - i - 1]` is 2. ......
Iteration 4:
    - `i` is 3, `arr[i]` is 4, and `arr[len(arr) - i - 1]` is 4. ......
Therefore, the final output of the function smallest_change([1, 2, 3, 4, 3, 2, 2]) is 2.

### Listing 7: HumanEval/142

```
def sum_squares(lst):
```

```
     result =[]
     for i in range(len(lst)): ## [STATE]i=[0, 1, 2][/STATE][STATE]range(len(lst))=[0, 1, 2][/
         STATE][STATE]len(lst)=[3][/STATE]
         if i %3 == 0: ## [CONDITION](i %3 == 0)=[True, False, True][/CONDITION]
[BRANCH]taken=[Y, N, Y][/BRANCH]
             result.append(lst[i]**2)
         elif i % 4 == 0 and i%3 != 0: ## [CONDITION](i % 4 == 0 and i%3 != 0)=[False, False,
             False][/CONDITION][CONDITION](i % 4 == 0)=[False, False, False][/CONDITION][
             CONDITION](i%3 != 0)=[False, False, False][/CONDITION][BRANCH]taken=[N, N, N][/
             BRANCH]
             result.append(lst[i]**3)
         else: ## [BRANCH]taken=[N, Y, N][/BRANCH]
             result.append(lst[i])
     return sum(result)

sum_squares([1,2,3])

Ground Truth:
(i%3) == 0: [True, False, False],
i%4 == 0 and i%3 !=0: [False, False, False]
else branch: [N, Y, Y]

MagiCoder-S:
Predicted output: 10
Analysis:
Iteration 1:
The index '0' is divisible by 3, so the square of the element at index 0 (1) is appended to
the result list.
Iteration 2:
The index '1' is not divisible by 3, so the element at index 1 (2) is appended to
the result list.
Iteration 3:
The index '2' is divisible by 3, so the square of the elementa at index 2 (3) is appended to
the result list.
The sum of the elements in the result list is 10.
```

### Listing 8: HumanEval/12

```
def longest(strings: List[str]) -> Optional[str]:
    if not strings: ## [CONDITION](not strings)=[False][/CONDITION][BRANCH]taken=[N][/BRANCH]
        return None
    maxlen = max(len(x) for x in strings)
    for s in strings: ## [STATE]s=['x', 'yyy'][/STATE]
## [STATE]strings=['x', 'yyy', 'zzzz', 'www', 'kkkk', 'abc'][/STATE]
        if len(s) == maxlen: ## [CONDITION](len(s) == maxlen)=[False, True][/CONDITION]
##[BRANCH]taken=[N, Y][/BRANCH]
            return s
longest(['x', 'yyy', 'zzzz', 'www', 'kkkk', 'abc'])

Ground Truth:
s=['x', 'yyy', 'zzzz']
output = 'zzzz'

DeepSeek-Coder-Instruct-6.7b
Predicted iutput: 'yyy'

Analysis:
......
Iteration 1:
    -The string 'x' has length 1, which is not the maximum length.
Iteration 2:
    -The string 'yyy' has length 3, which is the maximum length.
    -So, it returns 'yyy'.
Therefore, the final output of the function longest(['x', 'yyy', 'zzzz', 'www', 'kkkk', 'abc'
    ]) is 'yyy'.
```

### Listing 9: HumanEval/47

```
def fib4(n: int):
    results = [0, 0, 2, 0]
    if n < 4: ## [CONDITION](n < 4)=[False][/CONDITION][BRANCH]taken=[N][/BRANCH]
        return results[n]

    for _ in range(4, n + 1): ## [STATE]_=[4, 5][/STATE][STATE]range(4, (n + 1))=[4, 5][/STATE
        ][STATE](n + 1)=[6][/STATE]
        results.append(results[-1] + results[-2] + results[-3] + results[-4])
        results.pop(0)
    return results[-1]
fib4(5)
```

```
Ground Truth:
Output: 4

GPT-4-Turbo:
Predicted output: 10
The function fib4 is called with ( n = 5 ).
Inside the function, a list results is initialized to [0, 0, 2, 0].
The condition ( n < 4 ) is not satisfied, therefore the else branch is taken.
The loop iterates from 4 through 5 (since ( n = 5 )).
First Iteration (_ = 4):
    results.append(results[-1] + results[-2] + results[-3] + results[-4]) updates `results`
    to [0, 0, 2, 0, 4] and results.pop(0) removes the first element from `results`.
Second Iteration (_ = 5):
    results.append(results[-1] + results[-2] + results[-3] + results[-4]) updates `results`
    to [0, 2, 0, 4, 10] and results.pop(0) removes the first element from `results`.
After the loop completes, the function returns results[-1] which is 10 as the output.
```

Listing 10: HumanEval/160

```
def do_algebra(operator, operand):
    expression = str(operand[0])
    for oprt, oprn in zip(operator, operand[1:]): ## [STATE]oprt=['//', '*'][/STATE][STATE]
        oprn=[3, 4][/STATE][STATE]zip(operator, operand[1:])=[('//', 3), ('*', 4)][/STATE][
        STATE]operator=['//', '*'][/STATE][STATE]operand[1:]=[3, 4][/STATE]
        expression += oprt + str(oprn)
    return eval(expression)

do_algebra(['//', '*'], [7, 3, 4])
```

```
Ground Truth:
Output: 8

Gemini-1.5-Pro:
Predicted output: 2
Analysis:
The function do_algebra is called with (operator = ['//', '*'], operand = [7, 3, 4]).
Inside the function, a variable expression is initialized to '7'.
The loop iterates over the pairs of elements in operator and operand[1:].
First Iteration (oprt = '//', oprn = 3):
    expression += oprt + str(oprn) results in expression = '7//3'.
Second Iteration (oprt = '*', oprn = 4):
    expression += oprt + str(oprn) results in expression = '7//3*4'.
The function returns the result of evaluating the expression, which is 2.
```

Listing 11: HumanEval/11

```
def string_xor(a: str, b: str) -> str:
    def xor(i, j):
        if i == j: ##[CONDITION](i == j)=[True, False, False, False, False, False][/CONDITION]
            [BRANCH]taken=[Y, N, N, N, N, N][/BRANCH]
            return '0'
        else: ## [BRANCH]taken=[N, Y, Y, Y, Y, Y][/BRANCH]
            return '1'

    return ''.join(xor(x, y) for x, y in zip(a, b))
string_xor('111000', '101010')
```

```
Ground Truth:
(i==j) = [True,False,True,True,False,True]
if branch: [Y,N,N,N,N.N]
else branch: [N,Y,Y,Y,Y]
output: '010010'

Gemini-1.5-Pro
Predicted output: '010010'
Analysis:
The function string_xor takes two binary strings a and b as input and returns a new binary
string that is the result of XORing the two input strings. The function xor is a helper
function that takes two characters i and j as input and returns 0 or 1 values.
If `a` is '111000' and `b` is '101010', then the output of the `string_xor` function will be
'010010'
```

Listing 12: HumanEval/0

```
def has_close_elements(numbers: List[float], threshold: float) -> bool:
```

```
     for idx, elem in enumerate(numbers): ## [STATE]idx=[0, 1, 2, 3, 4, 5][/STATE][STATE]elem
         =[1.0, 2.0, 3.9, 4.0, 5.0, 2.2][/STATE][STATE]enumerate(numbers)=[(0, 1.0), (1, 2.0),
         (2, 3.9), (3, 4.0), (4, 5.0), (5, 2.2)][/STATE][STATE]numbers=[1.0, 2.0, 3.9, 4.0,
         5.0, 2.2][/STATE]
         for idx2, elem2 in enumerate(numbers): ## [STATE]idx2=[0, 1, 2, 3, 4, 5][/STATE]
## [STATE]elem2=[1.0, 2.0, 3.9, 4.0, 5.0, 2.2][/STATE]
             if idx != idx2: ## [CONDITION](idx != idx2)=True[/CONDITION]
##[BRANCH]taken=Y[/BRANCH]
                 distance = abs(elem - elem2)
                 if distance < threshold: ## [CONDITION](distance < threshold)=True[/CONDITION]
[BRANCH]taken=Y[/BRANCH]
                     return True

     return False
has_close_elements([1.0, 2.0, 3.9, 4.0, 5.0, 2.2],0.3)
```

**Ground Truth:**
idx2=[0,1,2,3,4,5,0,1,2,3,4,5];  elem2 = [1.0,2.0,3.9,4.0,5.0,2.2,1.0,2.0,3.9,4.0,5.0,2.2]
(distance<threshold) = [false,false,false,false,false,false,false,false,false,true]
output = true

**GPT-4-Turbo:**
Predicted output: True
**Analysis:**
The function uses two nested loops to compare each element with every other element in the list:
- The outer loop iterates through each element in the list, with idx as the index and elem as the element at that index.
- The inner loop also iterates through each element in the list, with idx2 as the index and elem2 as the element at that index.
For each pair of elements (elem, elem2) where idx is not equal to idx2 (ensuring they are distinct elements), the function calculates the absolute difference distance between elem and elem2. If this distance is less than the threshold, the function immediately returns True, indicating that there are close elements.
If no such pair is found after all comparisons, the function returns False.
**In this specific case, the input list is `[1.0, 2.0, 3.9, 4.0, 5.0, 2.2]` and the threshold is `0.3`. The function will return `True`.**

## Listing 13: HumanEval/98

```
def count_upper(s):
    count = 0
    for i in range(0,len(s),2): ## [STATE]i=[0, 2, 4][/STATE][STATE]range(0, len(s), 2)=[0, 2,
        4][/STATE][STATE]len(s)=6[/STATE]
        if s[i] in "AEIOU": ## [CONDITION](s[i] in "AEIOU")=[False, True, False]
        ## [/CONDITION][BRANCH]taken=[N, Y, N][/BRANCH]
            count += 1
    return count
count_upper('aBCdEf')
```

**Ground truth:**
(s[i] in "AEIOU")=[False, False, True]; output = 1

**CodeLlama-Instruct-13b:**
Predicted output: 1
**Analysis:**
The function count_upper is called with ( s = 'aBCdEf' ).
Inside the function, a variable count is initialized to 0.
The loop iterates from 0 through 4 (since ( len(s) = 6 ) and ( step = 2 )).
First Iteration (i = 0)
 - i is 0, which is not in the string 'aBCdEf'.
Second Iteration (i = 2)
    **- `i` is 2, which is 'B',** which is in the string 'aBCdEf'.
    **- The condition `s[i] in "AEIOU"` is true.**
    - Action: count += 1 updates count to 1.
Third Iteration (i = 4)
    - i is 4, which is 'E', **which is not in the string 'aBCdEf'.**
    **- The same condition `s[i] in "AEIOU"` is false.**

## Listing 14: HumanEval/73

```
def smallest_change(arr):
    ans = 0
    for i in range(len(arr) // 2): ## [STATE]i=[0, 1, 2, 3][/STATE][STATE]range((len(arr) //
        2))=[0, 1, 2, 3][/STATE][STATE](len(arr) // 2)=[3][/STATE]
        if arr[i] != arr[len(arr) - i - 1]: ## [CONDITION](arr[i] != arr[len(arr) - i - 1])=[
            True, False, False, False][/CONDITION][BRANCH]taken=[Y, N, N, N][/BRANCH]
            ans += 1
    return ans
```

Table 4: Comparison with REVAL and CODEMIND. *op* refers to output predition and *ic* refers to incremental consistency in (Chen et al., 2024). '-' means that the results of the model is not available in the official artifact.

| Models | CES | op(CodeMind) | op(REVAL) | ic(REVAL) |
|---|---|---|---|---|
| **CodeLlama-Instruct-7B** | 15.85% | - | 80.00% | 9.50% |
| *CodeLlama-base-7B* | 12.20% | - | 77.56% | 2.49% |
| **CodeLlama-Instruct-13B** | 13.41% | 75.85% | 79.51% | 3.36% |
| **DeepSeek-Coder-Instruct-6.7B** | 30.49% | 63.41% | - | - |
| **MagiCoder-S-6.7B** | 23.17% | 74.39% | 79.24% | 6.43% |
| **StarCoder2-15B** | 34.15% | 71.95% | 85.37% | 17.28% |
| **GPT-4-Turbo** | 47.56% | 82.93% | 87.80% | 42.72% |

```
smallest_change([1, 2, 3, 4, 3, 2, 2])

Ground Truth:
i = [0,1,2]
range((len(arr) // 2)) = [0,1,2]
(len(arr) // 2) = 3
output = 1

GPT-4-Turbo:
Predicted output: 1

Analysis:
The function smallest_change is called with the array [1, 2, 3, 4, 3, 2, 2].
The purpose of this function is to determine the minimum number of changes required to make
the array a palindrome.

The length of the array is 7, so the loop will iterate over the first half of the array,
which is the range from 0 to len(arr) // 2 (i.e., 0 to 3).

Iteration 1 (i = 0):
   - Compare arr[0] (which is 1) with arr[6] (which is 2).
   - They are not equal, so ans is incremented by 1.

Iteration 2 (i = 1):
   - Compare arr[1] (which is 2) with arr[5] (which is 2).
   - They are equal, so ans remains unchanged.

Iteration 3 (i = 2):
   - Compare arr[2] (which is 3) with arr[4] (which is 3).
   - They are equal, so ans remains unchanged.

Iteration 4 (i = 3):
   - Compare `arr[3]` (which is 4) with `arr[3]` (which is 4).
   - They are equal, so `ans` remains unchanged.

After completing the loop, the function returns the value of ans, which is 1. This indicates
that only one change is needed to make the array a palindrome.
```

## A.4    COMPARISON WITH ALTERNATIVE APPROACHES

In Table 4 we compare CES with REVAL and CODEMIND. Toward a fair comparison, we select 82 programs (containing 240 statements) that are used by all three approaches. From Table 4 we can observe that:

- CES achieves more pragmatic output prediction. Since CES discards suspiciously correct output predictions(§5.4) , the performance of LLMs on output prediction is 41.95% and 57.19% lower than that in CODEMIND and REVAL, respectively. It is infeasible for these two approaches to identify suspiciously correct output. REVAL only works on filtered statements thus it can not reason about the program as a whole. CODEMIND is capable of reasoning about code on the program level but it only requires LLMs to generate CoT along with the output which makes it hard to automatically evaluate the quality of the reasoning process.
- CES indicates higher consistency within the code reasoning tasks compared with REVAL. REVAL prompts LLMs separately and displays a very low incremental consistency. On the contrary, CES tracks the flow of the program within one prompt, and the correct output prediction with valid

reasoning process means that the model has consistent behavior in intermediate decision point prediction and output prediction. On average the consistency of CES is $187.54\%$ higher than the *ic* (incremental consistency) of REVAL, which reveals a more realistic consistency within the code reasoning task.

## A.5   BREAKDOWN OF THE EQUATION 1

After receiving the response of model $M$ for simulating the execution of program $P$ under inputs $I$, CES compares the ground truth with the predicted values for properties of individual statement in $l_j \in S_{loop}$, $c_j \in S_{condition}$, and $r_j \in S_{return}$ as below:

$$\forall\, l_j \in S_{loop} = \{l_1, \ldots, l_m\}$$
$$CES(M, P, I, I_{l_j}) = [\![\sum_{w=1}^{z} [\![M(P, I, I_{l_{j_w}}) = GT(P, I, I_{l_{j_w}})]\!] = z]\!] \tag{7}$$

$$\forall\, c_j \in S_{condition} = \{c_1, \ldots, c_n\}$$
$$CES(M, P, I, P_{c_j}) = [\![\sum_{w=1}^{z} [\![M(P, I, P_{c_{j_w}}) = GT(P, I, P_{c_{j_w}})]\!] = z]\!] \tag{8}$$

$$\forall\, r_j \in S_{return} = \{r_1, \ldots, r_k\}$$
$$CES(M, P, I, O_{r_j}) = [\![\sum_{w=1}^{z} [\![M(P, I, O_{r_{j_w}}) = GT(P, I, O_{r_{j_w}})]\!] = z]\!] \tag{9}$$

These three equations will be aggregated into Equation 1 for all the statements evaluated for code execution simulation.

## A.6   IN-DEPTH ANALYSIS ON RQ 4

The Venn diagrams of Figure 8 visualize the success cases per seven best-performing LLMs on CES, Bug Prediction, Bug Localization, and Bug Repair. To better understand the reasons for agreements and disagreements, we manually investigated instances where models (1) succeeded in CES and other bug-related tasks (instances inside the CES diagram) and (2) failed in CES but succeeded in bug-related tasks (instances outside of CES diagram), respectively. When we compare CES with any bug-related task, we can see that there are some unique problems in which the models can only produce correct predictions on CES or the bug-related task. This finding is consistent with the conclusion from the Kappa coefficients in Table 2.

GPT-4 Turbo and Gemini-1.5 Pro can succeed in all four tasks on the $33.75\%$ and $19.38\%$ problems in the benchmark, respectively. However, such cases could be very rare in less advanced LLMs. DeepSeekCoder-Inst-33b can consistently produce correct predictions on only $3\%$ of the problems, and such cases do not exist in CodeLlama-Inst-13b. By investigating the CoT reasoning of models in bug-related tasks, we observed that GPT-4 Turbo and Gemini-1.5 Pro simulate the program execution in their CoT, and use that to judge whether there is a bug in the code, localize the bug, and repair the bug. Figure 9 presents such an example from Gemini-1.5 Pro. In this Figure 9-a, `elif 1%3 != 0` can not handle the case where the index is a multiple of 4 and not a multiple of 3, which is specified in the natural language description in Figure 9-b. From Figures 9-c to 9-f, we can see that Gemini-1.5 Pro can correctly simulate the execution process of the test case. It determines that the program is buggy by predicting a different output (36) from the expected output (6). This example shows that LLMs that are more successful in bug-related tasks indeed consider code execution simulation in their reasoning process.

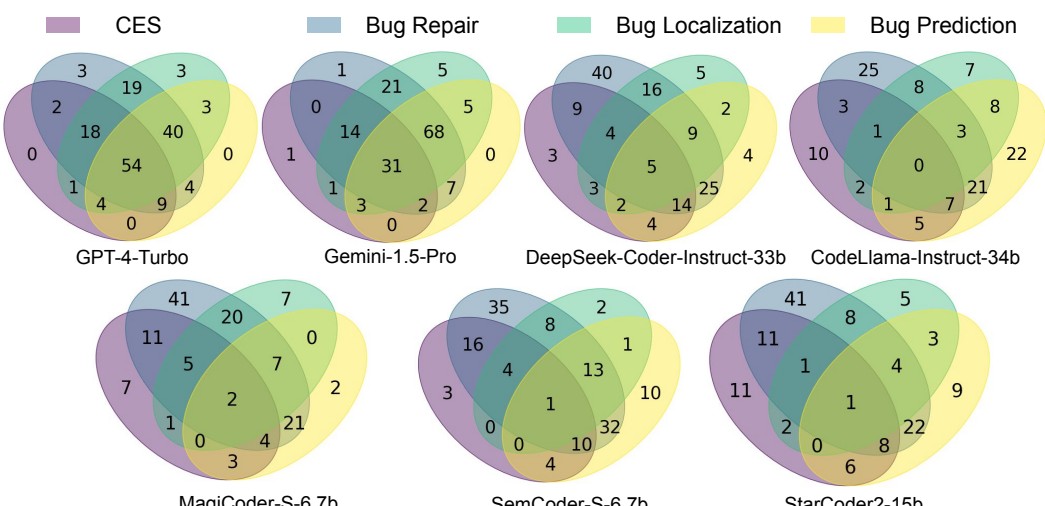

Figure 8: Correct predictions that representative LLMs make on CES, Bug Repair, Bug Localization, and Bug Prediction.

Figure 10 presents another example where Gemini-1.5 Pro makes the correct prediction in CES but fails in all three bug-related tasks. The bug of HumanEval/70 is in line 2, where `switch` is mistakenly initialized into `False` instead of `True`. As a result, given the input `strange_sort_list([1,2,3,4])`, the buggy code returns $[4, 1, 3, 2]$ instead of $[1, 4, 2, 3]$. Gemini-1.5 Pro correctly reasons about the loop property and the output of the buggy code (Figure 10-c) in CES. For the bug prediction (Figure 10-d), although it tries to simulate the execution of the given input, the simulation is incorrect. As a result, it concludes that the code is not buggy. Concerning bug localization (Figure 10-e) and bug repair (Figure 10-f), Gemini-1.5 Pro can successfully simulate the execution in its CoT. However, it hallucinates when localizing and repairing the bug, resulting in an incorrect performance. We believe that with a more formal reasoning approach concerning code execution that can be promptly validated, such as CES, LLMs will not fall into the trap of natural language hallucinations. As a result, when capable of performing code execution simulation, they can succeed in bug prediction, localization, and repair.

LLMs also could successfully predict, localize, and repair the bug while failing in the CES. Figure 11 shows such an example from the performance of GPT-4 Turbo on HumanEval/6 concerning different tasks. The buggy code is shown in Figure 11-a and the test input (`'(()(())((())))'`) is expected to return $[4]$. Due to the bug, however, the code returns $[3]$. From Figure 11-c, we can see that GPT-4 Turbo diverges from the ground truth execution from the beginning (prediction of loop variable `c` in the 7th iteration) and propagates the mistake through the execution, resulting in incorrect output prediction. From Figures 11-d to 11-f, we can see that GPT-4 Turbo is capable of predicting, localizing, and fixing the bug; however, the correct predictions are based on the incorrect understanding of the provided test case: in the CoT reasoning, it claims that the output of the test case matches the expected one on the buggy code. Despite the other examples, we also see that the code execution simulation in the CoT is either performed at a very high level or not at all. This example indicates that LLMs may neglect the test information or even incorrectly reason about the code execution, solely relying on the natural language specification to derive the results by chance, which can affect their trustworthiness.

From the examples above, we observe that LLMs are capable of following the step-by-step format to reason the execution process of the code when performing bug-related tasks, regardless of the correctness of the natural language CoT. However, CoT shortcuts and hallucinations may prevent LLMs from correctly performing code-related tasks. Therefore, we believe that combining a more systematic way, like CES, to represent the reasoning process in the pre-training / finetuning stage can improve LLMs on both code reasoning tasks and programming tasks that require awareness on the execution.

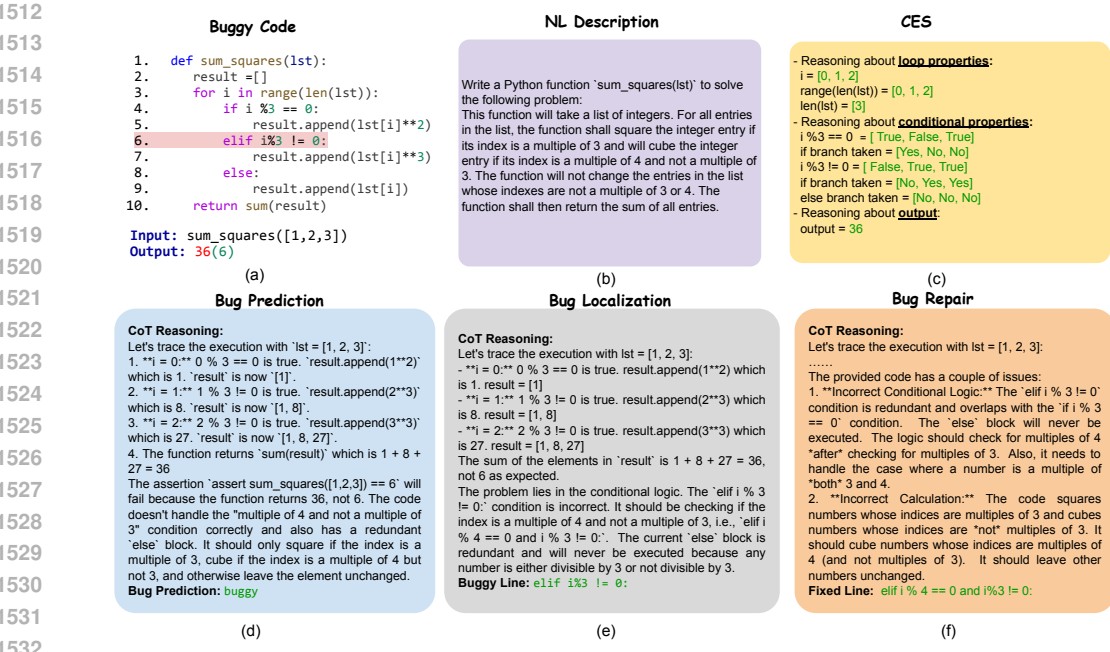

Figure 9: An example showcasing Gemini-1.5 Pro making correct predictions on CES (c), Bug Prediction (d), Bug Localization (e), and Bug Repair (f) for HumanEval/142.

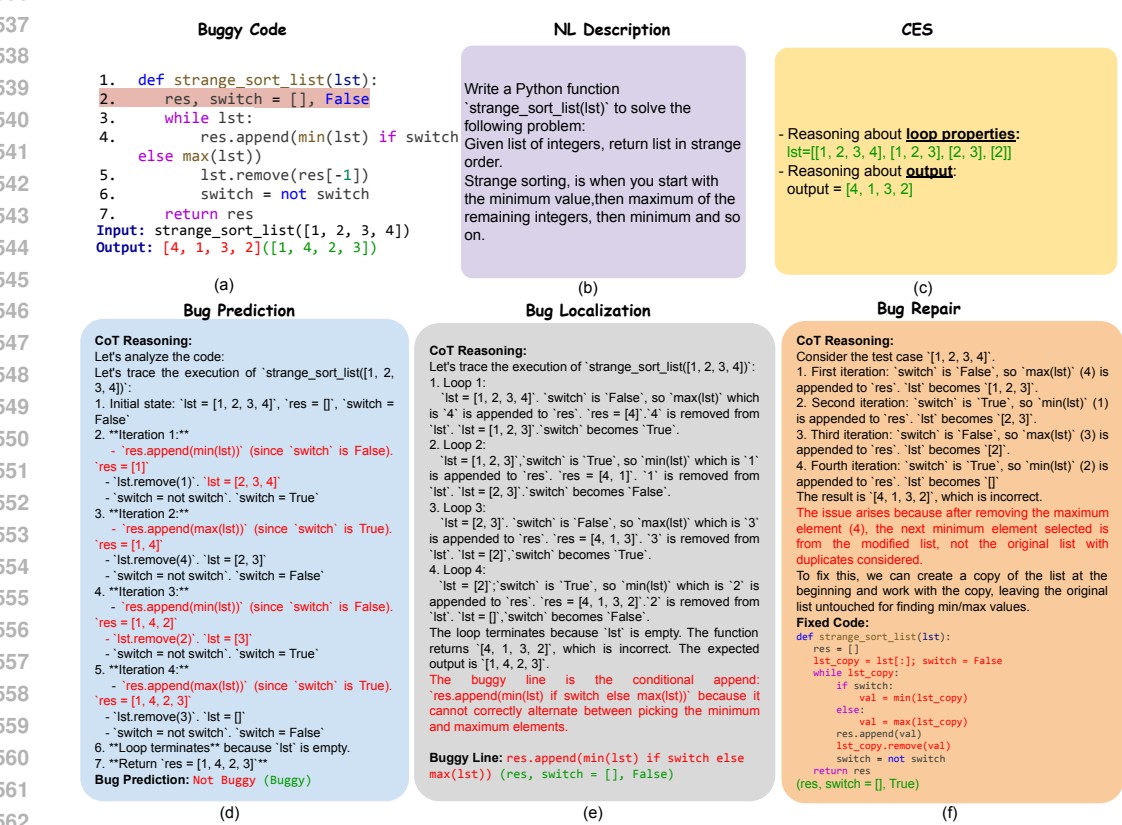

Figure 10: An example showcasing Gemini-1.5 Pro makes correct prediction on CES (c) but fails on Bug Prediction (d), Bug Localization (e), and Bug Repair (f) for HumanEval/70.

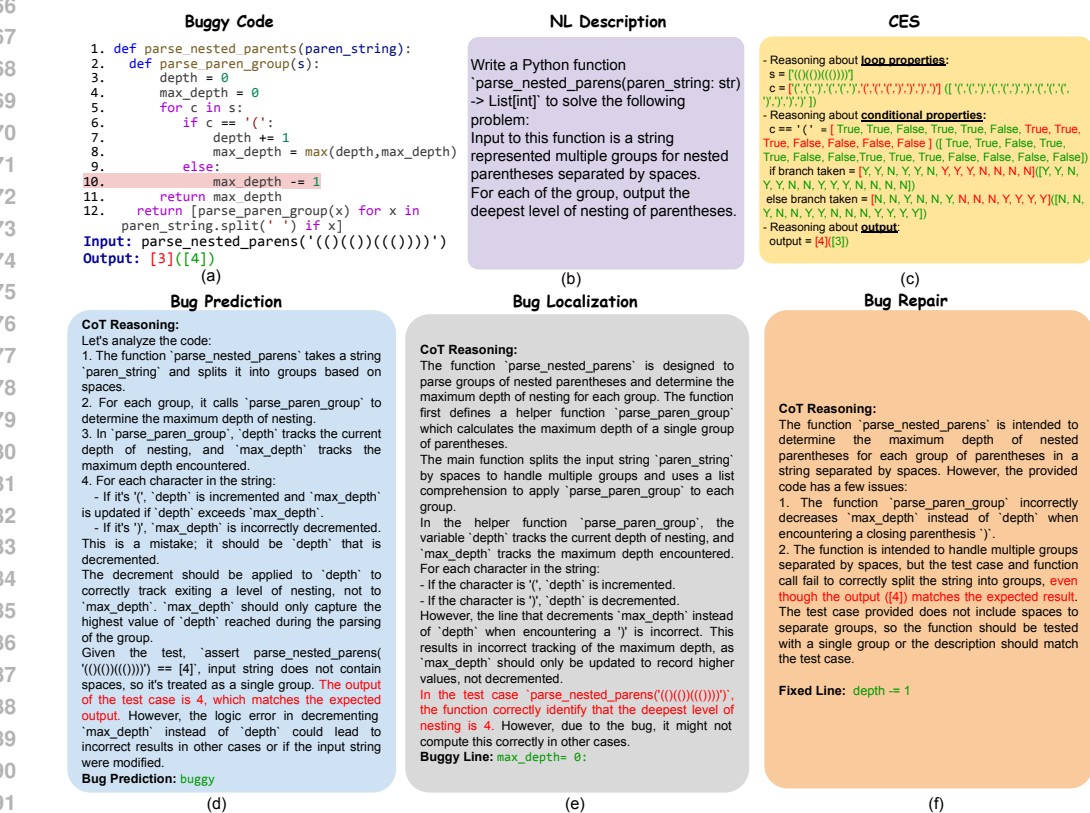

Figure 11: An example showcasing incorrect CES reasoning (c) by GPT-4 Turbo for HumanEval/6 problem (a), and correct Bug Prediction (d), Bug Localization (e), and Bug Repair (f) from GPT-4 Turbo for the same code. The specification for the functionality of HumanEval/6 is shown in (b).

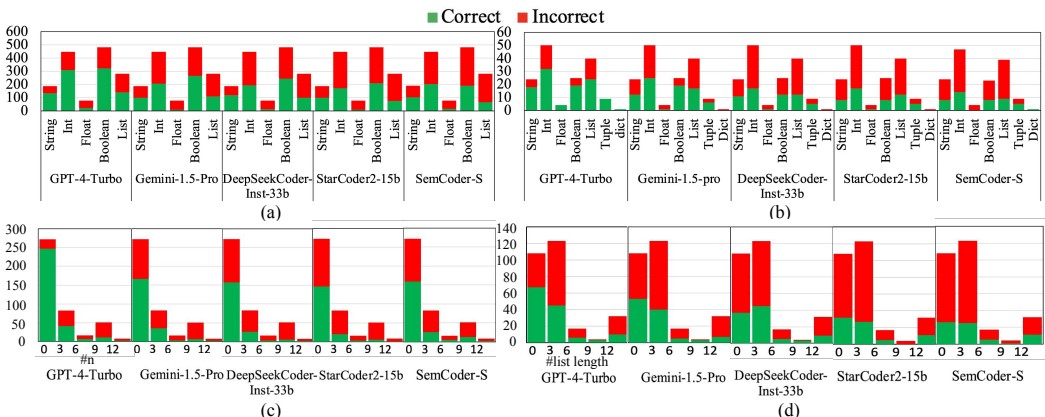

Figure 12: Number of correct/incorrect predictions on loop variables, loop iterables, conditional predicates (a), and return values(b) with different types. Impact of integer values (c) and loop length on the performances of LLMs.

## A.7 IMPACT OF VARIABLE TYPES

Figure 12, presents the number of correct and incorrect predictions on variables per different types. Here we collect all the variables that are predicted in CES, including loop variables, loop iterables, conditional predicates as well as return values, and categorize them into seven most common types in the programs: `String`, `Int`, `Float`, `Boolean`, `List`, `Tuple`, and `Dict`.

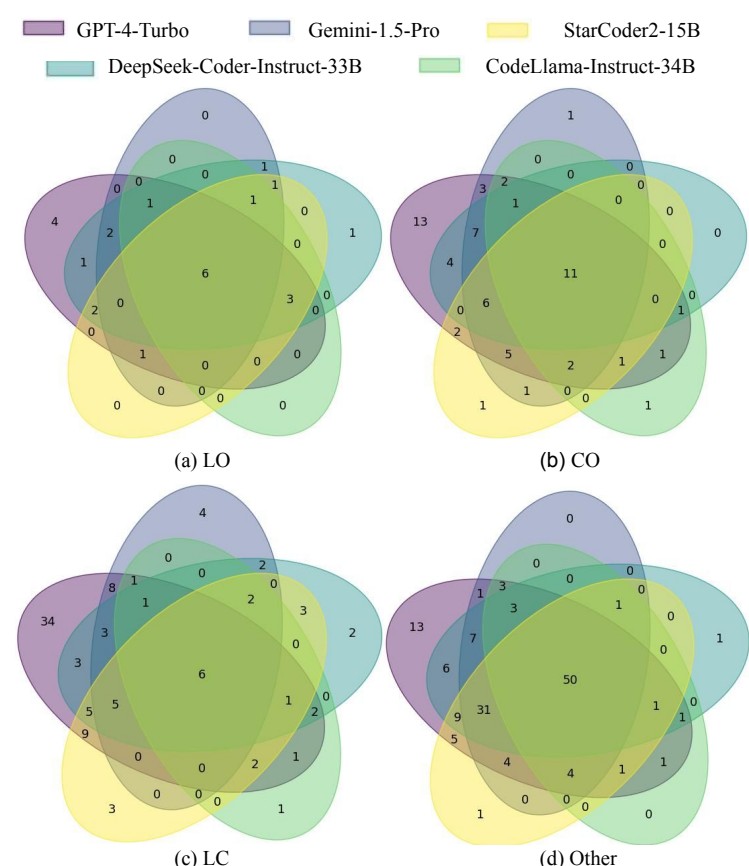

Figure 13: Comparison between the valid reasoning correct output prediction of representative LLMs.

From Figures 12-a and 12-b, we can see that predicting values of the float types is more challenging than other variable types for LLMs: the top five best-performing LLMs can only correctly predict 16.46% of them. We speculate that float is always associated with precise arithmetic computation, which has been proven to be one of the weaknesses of LLMs (Satpute et al., 2024). List is also very difficult for LLMs since it requires additional memory and recursion to understand every item inside it correctly. On average, LLMs can predict the value of 45.41% of variables in the type of list, which is lower than other primitive types (e.g., String, Int, and Boolean).

In Figures 12-c and 12-d, we further investigate how the absolute value of integers and the length of the lists impact the prediction of LLMs. Here we put all the ints and lists into different bins ($[0-2], [3-5], [6-8], [9-11], [12-+\infty]$) according to their values/lengths. We observe a similar trend in all models, that the percentage of correct predictions decreases as the integer value or list length grows, which indicates their negative impact on the code execution simulation performance of the LLMs.

## A.8 COMPARISON BETWEEN THE VALID REASONING CORRECT OUTPUT PREDICTION OF REPRESENTATIVE MODELS.

In Figure 13, we compare the valid reasoning process & correct output (also shown in Table 1) achieved by different models. From Figure 13-c, we can see that GPT-4 Turbo has 34 unique valid reasoning process & correct output compared to other models in the total 75 LC programs (programs with both loops and conditional statements, which shows its superiority in code execution reasoning. Similar trend can be found in LO programs (Figure 13-a), CO Programs (Figure 13-b) and Other (Figure 13-d).

**System Message**

You are given a piece of Python code and its input. You are asked to complete the following task: .......

[CODE]
```python
def sum_of_integer(N):
    sum_1 = 0
    for i in range(1,N+1): ## [STATE]i=??[/STATE]
[STATE]range(1,N+1)=??[/STATE] [STATE]N+1=??[/STATE]
        if i >= 1 and i <3:
        ## [CONDITION](i>=1)=??[/CONDITION]
        [CONDITION](i <3) = ??[/CONDITION]
        [CONDITION](i >= 1 and i <3)=??[/CONDITION]
        [BRANCH]taken=??[/BRANCH]
            sum_1 += 1
        ......
            sum_1 *= 4
    return sum_1
```
[/CODE]
[INPUT]
sum_of_integer(5)
[/INPUT]

[REASONING]
The function `sum_of_integer` is called with ( N = 5 ).
Inside the function, a variable `sum_1` is initialized to 0.
The loop iterates from 1 through 5 (since ( N = 5 )).
First Iteration (i = 1)
  - `i` is 1, which satisfies ( 1 <= i < 3 ).
  - The condition `i >= 1 and i < 3` is true.
  - Action: `sum_1 += 1` results in `sum_1 = 1`.
Second Iteration (i = 2)
  - `i` is 2, which still satisfies ( 1 <=i < 3 ).
  - The same condition is true.
  - Action: `sum_1 += 1` updates `sum_1` to 2 (`1 + 1 = 2`).
......

[/REASONING]

[PYTHON]
```python
def sum_of_integer(N):
    sum_1 = 0
    for i in range(1,N+1):
        ## [STATE]i=[1, 2, 3, 4, 5][/STATE]
        [STATE]range(1,N+1)=[1, 2, 3, 4, 5][/STATE]
        [STATE]N+1=[6][/STATE]
        if i >= 1 and i <3: ##
        [CONDITION](i>=1) = [True, True, True,
                True, True][/CONDITION].......
    return sum_1
```
[/PYTHON]

[OUTPUT]
24
[/OUTPUT]

**Adaptive In-context learning Examples**

[CODE]
```python
def count_vowels_consonants(input_string):
    vowels = 'aeiouAEIOU'
    vowel_count = 0
    consonant_count = 0
    for char in input_string:
## [STATE]char=??[/STATE]
[STATE]input_string=??[/STATE]
```
[/CODE]

[INPUT]
count_vowels_consonants("Hello, World!")
[/INPUT]

[REASONING]
The function `count_vowels_consonants` is called
with ( input_string = "Hello, World!" ).
Inside the function, a variable
`vowel_count` and `consonant_count` are initialized to 0.
......
[/REASONING]

[PYTHON]
```python
def count_vowels_consonants(input_string):
    vowels = 'aeiouAEIOU'
    vowel_count = 0
    consonant_count = 0

    for char in input_string:
##[STATE]char=['H','e','l','l',...,'d', '!'][/STATE]
[STATE]input_string=["Hello, World!"][/STATE]
        .......
```
[/PYTHON]
[OUTPUT]
(3, 10)
[/OUTPUT]

**Instruction**

You are given a piece of Python code and its input. You are asked to complete the following task:
1. Replace the "??" between [STATE] and [/STATE] in the comment...
2. Replace the "??" between [CONDITION] and [/CONDITION] in the comment...
3. Replace the "??" between [BRANCH] and [/BRANCH] in the comment with `Y` and `N`.
......

**Question**

[CODE]
```python
def longest(strings: List[str]) -> Optional[str]:
    if not strings: ## [CONDITION](not strings)=??
[/CONDITION][BRANCH]taken=??[/BRANCH]
        return None
    maxlen = max(len(x) for x in strings)
    for s in strings: ##
        if len(s) == maxlen:
            return s
```
[/CODE]

[INPUT]
longest(['x', 'y', 'z'])
[/INPUT]

**Answer**

[REASONING]
The function `longest` is called with ( strings = ['x', 'y', 'z'] ).
.......
[/REASONING]

[PYTHON]
```python
def longest(strings: List[str]) -> Optional[str]:
    if not strings: ## [CONDITION](not strings)=[True]
[/CONDITION]
[BRANCH]taken=[Y][/BRANCH]
        return None
......
```
[/PYTHON]

[OUTPUT]
None
[/OUTPUT]

Figure 14: The prompt template used in our study.

