# OpenReview forum: "Assessing Large Language Models for Valid and Correct Code Reasoning"
_ICLR.cc/2025/Conference — ICLR 2025 Conference Withdrawn Submission_

### Official Review · Reviewer_dsF5 · 2024-10-30

**Soundness:** 2
**Presentation:** 2
**Contribution:** 2
**Rating:** 3
**Confidence:** 4

**Summary:**

The paper introduces Code Execution Simulation (CES), a framework for evaluating code reasoning capabilities in large language models (LLMs). CES measures LLMs’ capacity to predict both output and intermediate program states, including loop variables, iterables, conditional predicates, and branches. Except for the prediction accuracy, CES can identify whether models have valid reasoning process and can determine their reasoning consistency by executing test cases with different prime path coverage. Through experiments on multiple LLMs, the authors find that while LLMs can achieve a high level of valid reasoning (82.32%), their reasoning strength remains inconsistent, often performing at random (55.59%) or weak (41.69%) levels.

**Strengths:**

- This paper presents a novel framework, Code Execution Simulation (CES), to assess code reasoning in LLMs. Code execution capability is an important aspect to evaluate LLMs.
- CES's design is diagnostic. By defining the notions of valid or invalid reasoning process, it can detect suspiciously correct output predictions under invalid reasoning.
- CES uses a novel reasoning consistency metric to benchmark LLMs' code reasoning abilities as strong, weak and random, by executing multiple tests per program with same or different prime path coverage.

**Weaknesses:**

- The primary weakness in this work lies in using code execution simulation as a measure of a model’s code reasoning abilities, which is debatable. While I agree with the authors' choice of evaluating reasoning validity (valid or invalid) and reasoning consistency as indicators of reasoning ability, as they reflect the model's understanding of code logic, the execution process itself requires substantial computational accuracy and strict adherence to instructions. For example, executing `a = b * c` demands multiplication skills, and executing `x = a[5]` requires precise indexing. The relationship between these computational abilities and code reasoning capabilities remains a research question. A model can easily compute `2 * 3`, yielding correct outputs in simple cases, but as inputs scale in complexity, the model's computational skills are challenged. However, this does not necessarily imply a lack of logical understanding or reasoning capability regarding the code’s logic. Thus, code execution simulation is inherently complex, and the authors do not sufficiently discuss this in the paper.
- The definition of the 'invalid reasoning process' is ambiguous. In Equation 4, the authors consider a compound property to be 'invalid' when it contains both correct and incorrect predictions. However, the example provided here involves the loop variable `o` and the loop iterable `zip(evens, odds)`. According to the definition given in Section 3.1, these two do not belong to the same property.
- The authors found in Section 5.5 that CES seems to have no correlation with other coding tasks, but they did not analyze the reasons for this. Is it because CES or bug-related tasks cannot represent the model's code reasoning ability, or do they focus on different aspects of reasoning ability? The authors also did not use other code comprehension tasks, such as code summarization, etc.
- It seems that there are several 'why' questions left unanswered in the evaluation. Why the predictions differed from ground-truth values? Why LLMs make suspiciously correct output predictions. The authors have relied solely on case analysis without providing quantitative data analysis.

**Questions:**

- Do the authors use the same prompts for different LLMs? How does the in-context learning examples affect the models' performances?
- To what extent does CoT (Chain of Thought) contribute to the results? Considering the hallucination phenomenon in LLMs , could the authors perhaps sample the output multiple times to observe the model's pass@k results?

---

> ### Author Response · Authors · 2024-11-27
> **Response to Reviewer dsF5 - part 1**
>
> **1-**  **[UNFAIR JUDGEMENT]** Using code execution simulation as a measure of a model’s code reasoning abilities
>
> We have added more results in the paper, discussing how the type and value of the variables impact the code execution simulation (Section A.7 in the Appendix). Predicting variable values, however, is only one aspect of code execution. To achieve a good performance in CES, LLMs should also predict correct branches or reason about how many times loop constructs are to be executed to predict a correct output ultimately. Our novel and systematic way of evaluating the reasoning process rules out cases where LLMs hallucinate or shortcut the reasoning, making it a fair and proper evaluating task.
>
> **2-**  **[FLAW]** The definition of the 'invalid reasoning process' is ambiguous
>
> We believe there is a misunderstanding about the example you mentioned. For the loop property, only the loop iterable can be a compound (3.1 Line 172). For example, in “**for o,e in zip(evens, odds),**”  the loop iterable “**zip(evens, odds)**” is a compound property because it has two sub-components: evens and odds. If the model correctly predicts “zip(evens, odds)” but mispredicts on “odds” or “evens,” then it will be an invalid reasoning process. “o” and “e” are loop variables, and they don’t belong to the same property as “zip(evens, odds)” per the **semantics of Python programming language**.
>
> **3-** In-depth analysis concerning RQ4.
>
> Thanks for your suggestion in the review. We added three subsections to the appendix with an in-depth analysis of RQ4 observations (Sections A.5-A.8 in the Appendix), which makes the paper stronger. In summary, we demonstrate more grounding evidence supporting our two hypotheses: Hypothesis 1- frontier LLMs that are more successful in bug-related tasks indeed consider code execution simulation in their reasoning process (Figure 9). They are still prone to natural language shortcuts and hallucinations that, despite correct code reasoning, prevent them from correctly performing bug-related tasks (Figure 10). Hypothesis 2- we show that LLMs, when they cannot correctly simulate the execution, still can perform well in bug-related tasks due to natural language shortcuts or out of luck (Figure 11).
>
> **4-** **[UNFAIR JUDGEMENT]** Code comprehension as a code reasoning tasks
>
> This paper focuses on code execution reasoning. There, indeed, can be other code reasoning tasks introduced to evaluate LLMs (a related work, CodeMind, does consider other code reasoning tasks such as specification reasoning). We focus on code execution reasoning, as we believe the current evaluation of LLMs for code reasoning is “**misleading**.” We propose a systematic way to identify hallucinations and shortcuts that result in false positives in existing evaluations. Our proposed technique is a fair and diagnostic way to replace existing misleading practices. Next, we plan to focus on other code reasoning tasks.
>
> **5-** **[UNFAIR JUDGEMENT]** Concerning your review, “The authors have relied solely on case analysis without providing quantitative data analysis.”
>
> Most of the analysis of the results in this paper is “**quantitative.**” We do answer all the “why” questions (Lines 455-497) through a meticulous, in-depth manual analysis, which seemingly is something that you do not appreciate. We respectfully ask for your suggested quantitative data analysis in addition to what has been performed in the paper. Kindly let us know what needs to be “**removed**” from the paper so that your suggestions “**fit into the page limit.**”
>
> **6-** Do the authors use the same prompts for different LLMs?
>
> We have designed a prompt template that all the models share. Before prompting each specific model, the prompt template will be modified with the best practices suggested by the model developers. For example, in prompting the DeepSeek-Coder, we use “###instruct” and “### response” to wrap the instruction. As another example, in prompting CodeLlama, we include “[INST]”,  “<<SYS>>”,  and “[/INST]” in the prompt.  Our artifacts are publicly available for further investigation.

---

> > ### Author Response · Authors · 2024-11-27
> > **Response to Reviewer dsF5 - part 2**
> >
> > **7-** Impact of in-context learning and CoT
> >
> > We report the ablation study results to demonstrate the effectiveness of the proposed adaptive in-context learning examples and CoT in (Appendix A.1). Experimental results in Table 3 show that adaptive in-context examples improve fixed in-context examples by 4.88%, on average. The CoT can improve the non-CoT setting by 4.11% concerning valid reasoning processes and correct output prediction. The Listing 2 in the revised manuscript presents an adaptive in-context example that improves DeepSeekCoder-Inst-33b on HumanEval/128, compared to the fixed in-context example in Listing 1. More details about the ablation can be found in Appendix A.1.
> >
> > **8-** Considering the hallucination phenomenon in LLMs , could the authors perhaps sample the output multiple times to observe the model's pass@k results?
> >
> > Thanks for your suggestion. We did not do this for the ablation study due to limitations on computing resources. Our experimental setup also uses temperature 0 to ensure minimal non-determinism.

---

### Official Review · Reviewer_N5Ju · 2024-11-01

**Soundness:** 2
**Presentation:** 3
**Contribution:** 2
**Rating:** 3
**Confidence:** 4

**Summary:**

The presented paper proposes a new method for assessing the capability of LLMs to predict intermediate states of program execution. The method picks specific/relevant parts of the program such as branching conditions and loops and prompts the LLM to predict the state at these lines during execution of the program with a specific input. The authors then analyze how well the LLM prediction aligns with the program state and use this to assess the capability of LLMs to correctly and consistently reason about program states and to diagnose at which point the LLM starts incorrect predictions.

**Strengths:**

- The paper is well-written and nicely structured. The figures and tables are well formatted and legible.
- The story is engaging and the tackled topic interesting.
- The proposed method promises improvements of previous work regarding the ability to pinpoint errors made by LLMs during reasoning at a lower inference cost.

**Weaknesses:**

1) Choice of predicted values

The proposed method compares to previous work that assesses the internal state of the program at other positions. It is not clear why exactly the proposed positions introduced in Sec 3.1. (branch taking, predicates, return value) specifically are the mainly relevant positions. The main argument appears to be that these are the most relevant values to detect and diagnose inconsistencies, and predicting further values would confuse the model.

It is possible that such assertions hold for the given dataset (or even more general programs) but I did not find any evidence pointing in this direction.

2) Confusing Definitions of valid/invalid reasoning

a) No correspondence to "consistency"

The authors mark any reasoning as invalid (Sec 3.3.) if an intermediate state (i.e. predicate) is incorrectly predicted but the consequence is predicted correctly (i.e. the branch-taking based on the predicate). This appears to not accurately capture whether the _reasoning_ was indeed wrong, since the intermediate state and consequence could still be _consistent_ (i.e. for "if p or not p:", it does not matter what is predicted for p to correctly predict that the branch is taken) and thus not represent a case of incorrect reasoning. It could or could not be that in the given dataset the introduced invalid reasoning always constitutes incorrect reasoning, but such evidence is missing from the paper.

b) Incorrect outputs are valid reasoning

The definition of "valid reasoning" includes (by definition) all instances where the model outputs incorrect output. This naming is confusing at best, since I would not expect that incorrect instances can constitute valid reasoning. As already mentioned in 2) this is due to a lack of evaluation of _consistency_ which I would consider indicative of reasoning.

3) Weak performance at CES may imply subpar benchmark design

In Sec 5.2 the authors mention many cases of "suspiciously correct" outputs based on natural language reasoning and inconsistent with the produced code reasoning. My interpretation of this would be that the presented code evaluation is potentially unnatural and confusing to the language model and thus artificially reduces performance, where free-form reasoning in natural language allows the models to correctly derive a result. Interesting counter-evidence for such an interpretation would be that models also often override correctly reasoned code states with incorrect (i.e. biased through function names) natural language reasoning results.

Similarly in Sec 5.5. weak correlation of models in CES and other related program understanding tasks do not necessarily imply that models are subpar reasoners, instead it could also imply that CES is not a format in which models can effectively express their code understanding.


The following are some smaller points that left me confused after reading:
- In Sec 5.1. the authors mention that there is no control for the coverage of test cases on programs. This appears weird, it would be interesting to somehow establish a controlled experiment for different path coverage. The detailed Figure 6 partially makes up for this.
- Figure 7 is very difficult for me to parse, especially the legend, but also the choice of chart format, respective the choices of grouping (it might make more sense to overlay models on triangular LO, CO, LC chart?)
- Figure 8: The instruction to the model reads "You are given a piece of Python code and its _output_" while the model is clearly given _input_ below. I hope this is a typo, otherwise it might have implications for the presented numbers.
- In Sec 5.2. "Invalid Reasoning" it reads "[…] LLMs with good performance in valid reasoning also make more invalid reasoning". This seems contradictory since reasoning is either valid or invalid, and the sum of it should be constant - thus increasing one would necessarily decrease the other. Please do clarify what is meant by this statement.

**Questions:**

Please provide a short statement or clarification to the points raised above.

---

> ### Author Response · Authors · 2024-11-27
> **Response to Reviewer N5Ju- part 1**
>
> **1-** **[UNFAIR JUDGEMENT]** Choice of predicted values
>
> We have justified our choice in Section 3. Generally speaking, a program consists of four main categories of statements: assignment, condition, loop, and return. **We have included all categories except assignment statements.** These are the main programming points that, depending on the inputs, identify the execution flow of the program. Including assignment statements in our initial experiments resulted in a poor performance of the models. Furthermore, loop properties, conditional statements, and the return value identify the start or end of basic blocks in the program control flow graph and **can capture mispredictions in the assignment statements inside the block**. Our formalizations are generic and **PL-agnostic**. For example, conditional statements in Java include if/else statements and switch/cases. In Python, it could be if/else/elif/match-case. Our code has been released publicly, and you can check out our implementations. If you question our design decisions and reject the paper based on that, please suggest an alternative. Otherwise, this is an unfair judgment.
>
> **2-** **[UNCLEAR review and UNFAIR JUDGEMENT]** Confusing Definitions of valid/invalid reasoning: no correspondence to "consistency"
>
> There is indeed no notion of consistency in evaluating valid or invalid reasoning processes. Consistency in the literature is defined as a quality measurement between different prompts of the model, not its performance in one prompt. We also explain how CES works with your given example, and hopefully, it clarifies the strength of the proposed technique for you.
> The predicate of “**(p or not p)**” consists of two sub-predicates of (1) **p** and (2) **not p**. CES evaluates the values of each sub-predicate, the predicate, and the branch. Suppose that the ground-truth value for p is True, making the ground-truth values for the first sub-predicate True and the second sub-predicate False. Considering the permutations for sub-predicates, predicate, and the branch, 16 possible outcomes concerning the model’s response are listed below. CES rules out the invalid reasoning processes due to Equation 4 (cases 2,3,4,10,11,12) and Equation 3 (cases 5,6,7,8,9). The remaining cases, although per human logic are invalid (e.g., True or True cannot be False), cannot be captured by CES, as it cannot understand the logical operators. In these cases, if the output prediction is incorrect, it will be discarded anyway. If the output prediction is correct, they will be ruled out as invalid per Equation 2. So, in the end, CES takes care of all cases.
> **Please also note that, in practice, people only count the valid reasoning process and correct output prediction as the model's success. We have studied invalid reasoning process cases to understand better how LLMs perform the code execution simulation.**
>
> <sub-predicate1,sub-predicate2,predicate,branch>
> 1- <True,False,True,True>
> 2- <True,True,True,True>: invalid (Equation 4)
> 3- <False,False,True,True>: invalid (Equation 4)
> 4- <False,True,True,True>: invalid (Equation 4)
> 5- <True,False,False,True>: invalid (Equation 3)
> 6- <True,True,False,True>: invalid (Equation 3)
> 7- <False,False,False,True>: invalid (Equation 3)
> 8- <False,True,False,True>: invalid (Equation 3)
> 9- <True,False,True,False>:  invalid\lor (Equation 3)
> 10- <True,True,True,False>: invalid (Equation 4)
> 11- <False,False,True,False>: invalid (Equation 4)
> 12- <False,True,True,False>: invalid (Equation 4)
> 13- <True,False,False,False>: discarded if output prediction is incorrect or ruled out by Equation 2 as invalid
> 14- <True,True,False,False>: discarded if output prediction is incorrect or ruled out by Equation 2 as invalid
> 15- <False,False,False,False>: discarded if output prediction is incorrect or ruled out by Equation 2 as invalid
> 16- <False,True,False,False>: discarded if output prediction is incorrect or ruled out by Equation 2 as invalid

---

> ### Author Response · Authors · 2024-11-27
> **Response to Reviewer N5Ju- part 2**
>
> **3-** Confusing Definitions of valid/invalid reasoning: incorrect outputs are valid reasoning
>
> Please note that **starting from the abstract**, we use the term “**invalid reasoning process**” to differentiate between variable predictions and reasoning processes. Models may make a mistake in predicting intermediate values. As long as they propagate the mistake, this is a valid reasoning process (although the outcome will be discarded because the predictions were incorrect). However, we show that LLMs cheat and try to cover for their mistakes with shortcuts or hallucinations. We systematically identify such cases as “invalid reasoning processes” by LLMs. The notion of the reasoning process and systematically evaluating it to be valid or invalid is, in fact, the notable contribution of this work. We understand that due to the novelty of the concept and term, it can be confusing at the beginning. Although wordy, we revised the manuscript and added “**process**” when discussing valid or invalid reasoning to avoid confusion. **Please let us know if you have better suggestions**.
>
> **4-** **[FLAW and UNFAIR JUDGEMENT]** Concerning your review, “Interesting counter-evidence for such an interpretation would be that models also often override correctly reasoned code states with incorrect (i.e. biased through function names) natural language reasoning results.”
>
> The original submission **does contain** this specific example (Listing 9 in the Appendix) and a reference to it in the main text (Lines 474-477), demonstrating the “**counter-evidence**” you were looking for.
>
> **5-** Weak performance at CES may imply subpar benchmark design
>
> We respectfully disagree, and the new experiments and results support our claim. Our newly added discussions from an in-depth analysis of RQ results in Section A.6 (Appendix) could resolve this concern. In summary, we show that in most cases where the model could not reason about the code execution, the success in bug-related tasks is due to incorrect hallucinations or shortcuts. We also observe that in the majority of the cases where the models correctly predicted, localized, or repaired the bug, they did try to simulate the execution of the program as part of their reasonings. Furthermore, we added the results of the ablation study, demonstrating the impact of different design choices in the CES task, confirming that CES is well-designed and fairly evaluates models for code reasoning. Please also consider that we are proposing a systematic method for better code reasoning and overcoming **important limitations** of prior work.
>
> **6-** Smaller points
>
> Thanks for your feedback. We have fixed the typos, added clarifying texts, and updated Figure 7 to make it more legible.

---

> ### Comment · Reviewer_N5Ju · 2024-11-28
>
> I thank the authors for their extensive rebuttal. The provided comments do not address my concerns satisfyingly, I provide further reasoning for this below. Overall, I think the presented method misses a big opportunity in not measuring whether CES is the appropriate format to elicit code reasoning in language models.
>
> ### 1 **Choice of predicted values**
>
> Indeed, I question the choice of excluding assignment statements. I suggest the authors properly explore this alternative and provide experimental results for this ablation.
>
> As the authors correctly point out, the set of statements covered by CES *can* capture mispredictions in assignments. What makes the authors so sure that it does so sufficiently?
>
> They further argue that predicting the value of assignment statements "results in poor performance" - a proper experiment for this would be an interesting ablation and provide strong evidence for the claimed superiority of the chosen subset of statements. The provided reference in Section 3 merely generally argues that long contexts deteriorate model performance, but it is not clear that this applies here and outweighs the potential positive effect of more detailed insight in predicted values.
>
> ### 2/3 **Confusing definition of valid/invalid reasoning**
>
> Please clarify what you mean by saying "it will be discarded anyway".
> > if the output prediction is incorrect, it will be discarded anyway.
>
>  I read in your abstract that models follow 80% "valid reasoning" with 62% of that (50% in total) incorrect output predictions. This appears misleading to me, because the label "valid reasoning" would imply that you inspect the reasoning and consider it valid, instead you label anything as valid that is incorrect. Mentioning this non-inspected incorrect prediction as the result of "valid reasoning" in, among other places, the abstract seems to me anything but "discarding" the result.
>
>  I suggest the authors either inspect the validity of reasoning also for incorrect predictions or alternatively refer to it as, e.g. "unchecked reasoning".
>
> ### 4 **Confusing definition of valid/invalid reasoning**
>
> My concern is the following: The formatting of CES appears very restrictive and unnatural and could itself cause the model to generate incorrect results. Other formats could be easier to follow and allow the model to correctly infer the intermediate states and output of a function. The authors do not ablate on this format, such an ablation would help significantly strengthen the claims in the paper.
>
> The referred Listing 9 appears to be unrelated to my concern and in Listing 11, referenced in Lines 474-477, the model generates incorrect intermediate states but correctly infers the output in its free-form reasoning (albeit arguably through a "shortcut"). This is exactly not the counter-evidence I was asking for, which would be that the model correctly generates intermediate states for a buggy program, then overrides these states because of its natural language reasoning, being biased by the program name. Since this coincides with an incorrect output prediction, I assume such cases where "discarded" by the authors and not further investigated?
>
> ### 5 **Weak performance at CES may imply subpar benchmark design**
>
> The new experiments in Appendix A.6 strengthen my suspicion that CES is not ideal for models to express code reasoning. As can be seen in Figure 8, stronger models (GPT-4, Gemini 1.5 Pro) appear to consistently perform well in bug localization, prediction and repair, only that there is no correlation to CES. Meanwhile weaker models appear to perform almost randomly across all tasks. I don't see the clear benefit of CES here.
>
> The example in Figure 10 appears highly unrepresentative of Gemini 1.5 Pro behavior, it is the only instance where it passes CES but fails on all bug related tasks (according to Figure 8).
>
> Figure 11 is indeed interesting but it is also unclear if this is representative. To provide more convincing proof that this is indeed representative, the authors could manually check all 40 instances and report the percentage of cases where GPT-4 uses hallucination to resolve bug localization, repair and prediction.

---

> ### Author Response · Authors · 2024-11-30
> **Response to Reviewer N5Ju - part 1**
>
> 1- Choice of predicted values
> In CES, we use statements that mark the beginning/end of basic blocks in the program to capture the mispredictions of assignment statements. For example, in Figure 5-a, if the model fails on the assignment statement ‘m=e’ in line 5, then this error will be captured in the return statement ‘return m’ in line 6.  Similarly, in Figure 1-b, if mistakes happen in line 9 and line 10, then CES can use line 11 and line 8 to catch them.
> We also realize that the format of CES can be not that “natural” for LLMs compared to free-form natural language, and to this end, we carefully design our prompt (more details can be found in Appendix A.1) and ask the model to predict important decision points in the program instead of everything. Anyway, we believe your suggestion is valuable, we will definitely explore alternatives to include more statements into CES in our future work.
>
> **2-**  **[Unfair Judgement]** 2/3 Confusing definition of valid/invalid reasoning
> We mentioned “it will be discarded anyway” **only for the special case “(p or not p)” exclusively.** As we emphasized in Section 3.3 and the previous response, the notion of reasoning validity enables CES to identify (1) valid reasoning process and correct output, and (2) invalid reasoning process and correct output. If the output predictions of case 13, 14, 15, and 16 are correct, they will belong to something other than these two categories we are more interested in. Meanwhile, these four cases may contain some invalid human logic (e.g., True or True cannot be False) that CES can not capture. Also given the low probability of running into this special case in the real-world program, we “discard it anyway” in the discussion in the previous response.
>
> In CES, we check the validity of all the reasoning processes, and our response to Reviewer QzKa provides a more detailed explanation. Again, in our study, there is no “unchecked reasoning,” and all the reasoning processes are checked using Equations 2, 3, and 4. Please revisit Section 3.3 and let us know if any reasoning process is not inspected in our definition.

---

> > ### Author Response · Authors · 2024-11-30
> > **Response to Reviewer N5Ju - part 2**
> >
> > **3-** Confusing definition of valid/invalid reasoning
> > According to your description, “the model correctly generates intermediate states for a buggy program, then overrides these states because of its natural language reasoning”, we present another example from Gemini-1.5-Pro on HumanEval/104:
> >
> > **Buggy Code:**
> > ```python
> > def unique_digits(x):
> >     odd_digit_elements = []
> >     for j, i in enumerate(x):
> >         if all (int(c) % 2 == 1 for c in str(i)):
> >             odd_digit_elements.append(i)
> >             odd_digit_elements.append(j)
> >     return sorted(odd_digit_elements)
> > ```
> >
> > **Input:**
> > ```
> > unique_digits([15, 33, 1422, 1])
> > ```
> > **Ground truth output:**
> > ```
> > [0, 1, 1, 3, 15, 33]
> > ```
> > **CES (intermediate state):**
> > ```python
> > def unique_digits(x):
> >     odd_digit_elements = []
> >     for j, i in enumerate(x): ## [STATE]j=[0, 1, 2, 3][/STATE][STATE]i=[15, 33, 1422, 1][/STATE][STATE]enumerate(x)=[(0, 15), (1, 33), (2, 1422), (3, 1)][/STATE][STATE]x=[15, 33, 1422, 1][/STATE]
> >         if all (int(c) % 2 == 1 for c in str(i)): ## [CONDITION](all (int(c) % 2 == 1 for c in str(i)))=[True, True, False, True][/CONDITION][BRANCH]taken=[Y, Y, N, Y][/BRANCH]
> >             odd_digit_elements.append(i)
> >             odd_digit_elements.append(j)
> >     return sorted(odd_digit_elements)
> > ```
> > **Natural language reasoning:**
> >
> > (1). Initialization:
> >    - `odd_digit_elements` is initialized as an empty list.
> > (2). Loop 1:
> >    - `j = 0`, `i = 15`
> >    - `str(i)` becomes "15".
> >    - The condition `all(int(c) % 2 == 1 for c in str(i))` checks if all digits in "15" are odd.
> >    - Since '5' and '1' are odd, **the condition is `False` (because not all digits are odd).**
> >    - **The `if` block is skipped.**
> >
> > (3).Loop 2:
> >    - `j = 1`, `i = 33`
> >    - `str(i)` becomes "33".
> >    - The condition checks if all digits in "33" are odd.
> >    - Since '3' is odd, the condition is `True`.
> >    - `odd_digit_elements.append(i)` adds 33 to the list.
> >    - `odd_digit_elements.append(j)` adds 1 to the list.
> >    - `odd_digit_elements` is now `[33, 1]`.
> >
> > (4). Loop 3:
> >    - `j = 2`, `i = 1422`
> >    - `str(i)` becomes "1422".
> >    - The condition checks if all digits in "1422" are odd.
> >    - Since '4' and '2' are even, the condition is `False`.
> >    - The `if` block is skipped.
> >
> > (5). Loop 4:
> >    - `j = 3`, `i = 1`
> >    - `str(i)` becomes "1".
> >    - The condition checks if all digits in "1" are odd.
> >    - Since '1' is odd, the condition is `True`.
> >    - `odd_digit_elements.append(i)` adds 1 to the list.
> >    - `odd_digit_elements.append(j)` adds 3 to the list.
> >    - `odd_digit_elements` is now `[33, 1, 1, 3]`.
> >
> > (6). Return:
> >    - `sorted(odd_digit_elements)` sorts the list to `[1, 1, 3, 33]`.
> >    - The function returns `[1, 1, 3, 33]`.
> >
> > Predicted output: [1, 1, 3, 33]
> >
> > In this example, Gemini-1.5-Pro can correctly predict all the loop variables, loop iterables, conditional predicates and branches. In the natural language reasoning, it incorrectly predicts the conditional predicate and the branch taking in the first iteration of the loop (**bolded**) and overwrites the correct prediction in CES. Finally, it mispredicts the output.
> >
> > Again, all the reasoning processes are inspected, and none are discarded simply because of the incorrect output prediction.

---

> > > ### Author Response · Authors · 2024-11-30
> > > **Response to Reviewer N5Ju - part 3**
> > >
> > > **4-** Weak performance at CES may imply subpar benchmark design
> > >
> > > Figure 8 shows that although stronger models consistently have better performances on Bug Prediction, Bug Localization, and Bug Repair compared to weaker models, there is still a low correlation between CES and bug-related tasks, which raises the following question: to what extent do LLMs attend to the execution of the code when they are dealing with bug-related tasks? In Appendix A.6, we use some examples to show that LLMs may hallucinate, make shortcuts in the reasoning process, or even simply rely on the natural language specification to perform bug-related tasks.
> > >
> > > Indeed, Figure 10 is the only instance where it passes CES but fails on all bug-related tasks for Gemini-1.5-Pro. According to Figure 8, there can be more similar cases for weaker LLMs. The point of Figure 10 is to show that, even for a stronger LLM, it may still fall into the trap of natural language hallucinations, which may finally result in incorrect predictions on bug-related tasks. Therefore, we argue that teaching LLMs with a more formal reasoning approach concerning code execution can be an alternative to reduce such natural language hallucinations and further improve LLMs’ performances on bug-related tasks.
> > >
> > > Following your suggestion, we manually checked the 40 instances in GPT-4-turbo, where it can successfully carry out bug-related tasks but fails on CES. We find 17 instances (42.50%) where the model uses hallucination or merely natural language specification to correctly predict, locate, or fix the bug.
> > >
> > > Problem id of the 17 instances: HumanEval/36, HumanEval/59, HumanEval/96, HumanEval/21, HumanEval/73, HumanEval/1, HumanEval/20, HumanEval/12, HumanEval/109, HumanEval/80,
> > > HumanEval/116, HumanEval/41, HumanEval/6, HumanEval/161, HumanEval/154, HumanEval/55, HumanEval/134. More details can be found in our artifact.

---

> > > > ### Comment · Reviewer_N5Ju · 2024-12-02
> > > >
> > > > I thank the authors for their extensive rebuttal. I have thoroughly investigated the posted comments and remain at my given score. The reason is that my major concerns are not addressed adequately.
> > > >
> > > > I acknowledge that the provided example in the last response meets my demands of showcasing correct CES reasoning and incorrect NL reasoning co-occuring, and measuring that roughly half the GPT-4 instances that fail on CES but succeed fixing bugs use hallucination to fix bugs.
> > > >
> > > > However my remaining concerns are too strong to recommend acceptance for this paper:
> > > >
> > > > - The paper does not contain, and the authors did not provide, concrete, quantitative justification for their choice of evaluated statements, nor their choice of reducing outputs into a single prompt, which I suspect significantly reduces LM capability and draws into question the discovered dissonance between bug-related tasks and code reasoning (i.e. in the other half of the above GPT-4 cases, CES reasoning fails and natural language reasoning _correctly and without hallucination_ identifies bugs).
> > > > - According to the authors, only the combination of invalid reasoning and valid outputs or valid reasoning and valid outputs are relevant. However throughout the paper, other combinations are presented and highlighted as well. The definition of "valid reasoning" has been questioned by me and reviewer QzKa and the fundamental issue remains that only "correct prediction of X but incorrect prediction of subcomponents of X" is "invalid", whereas everything else is "valid", which makes no intuitive sense and appears misleading. Since the discrimination between reasoning and outputs is claimed as a major contribution of the work, it should be clear and useful to readers.
> > > >
> > > > Finally the paper is still not adequately formatted for publication and the rebuttal leaves me unsure about whether the authors would be able to correct this formatting for a camera-ready submission. Among other issues, the presentation of Figure 7 has been criticized by me and Reviewer ehpE and only minimally improved in readability, and the addition of the word "process" in the revised paper has introduced plenty of broken grammar, while IMO not improving clarity at all.

---

> > > > > ### Author Response · Authors · 2024-12-02
> > > > > **Response to Reviewer N5Ju**
> > > > >
> > > > > Thanks for acknowledging that our experiments in the paper address most of your concerns. It seems that you have decided to reject this paper from the beginning, and we found your review biased based on that mindset.
> > > > >
> > > > > - Regarding your first bullet point, we believe your bias caused you to completely ignore this work's contributions. Our motivating/illustrative example in Figure 1 shows what happens if you prompt the models separately concerning the same code. In many studies, a counter-example is enough for not wasting computing resources to go for an ablation! Comparing a well-thought-out approach with several optimizations in the prompts with a naive ablation that asks the model for everything is counter-intuitive.
> > > > >
> > > > > - We have tried to answer you ***with examples, referring to sound formulations and peer-reviewed research papers***. You keep accusing our proposed technique without such evidence. So, we are asking you to clarify how including the statements in the prompt or having multiple prompts helps your concern of "I suspect significantly reduces LM capability and draws into question the discovered dissonance between bug-related tasks and code reasoning" being resolved. Your text is very unclear.
> > > > >
> > > > > - Please quote us where we mentioned "**only** the combination of invalid reasoning and valid outputs or valid reasoning and valid outputs are relevant."
> > > > >
> > > > > - Your statement of "correct prediction of X but incorrect prediction of subcomponents of X" is "invalid", whereas everything else is "valid" is **wrong**. What you have mentioned only reflects Formula (4) and not the other two. We are truly overwhelmed that you keep mentioning an incorrect understanding of the paper that is not aligned with the formalization we have provided.
> > > > >
> > > > > - When you say "which makes no intuitive sense and appears misleading," support your claim with at least a counter-example. We believe our answer to your tricky question of if(p or not p) clarifies this. Use that example to show that we are incorrect and misleading.
> > > > >
> > > > > - Please let us know what further concrete changes to Figure 7 are required to improve the presentation. Your feedback can certainly help.
> > > > >
> > > > > We also have a question: Identify why a new task that evaluates LLMs from a new aspect essential in programming (having LLMs simulate the entire programming stack) is misleading. Several code execution reasoning approaches have already been published in top venues. We are showing their important limitations and how "those research can be misleading." We are sorry that you think including a counterintuitive ablation is more important than showing that existing research is misleading.

---

### Official Review · Reviewer_QzKa · 2024-11-03

**Soundness:** 3
**Presentation:** 3
**Contribution:** 3
**Rating:** 6
**Confidence:** 3

**Summary:**

The paper proposes a Code Execution Simulation (CES) task to evaluate how well current language models understand and are able to execute code. The tasks is simulated the execution of a program by following the execution trace and producing the intermediate states. They introduce two aspects that go beyond code execution results correctness: checking if the simulated execution trace deviates from the correct program execution trace, and identifying situations where the model gets the right answer through questionable means. They also investigate how consistently these models perform with different test cases covering different execution paths. They find that LLMs still struggle with execution, especially for tasks that are more control flow sensitive.

**Strengths:**

- The paper provides a very thorough investigation of LLMs' capability for code execution. They not only provide a reasonable framework to define strong or weak code execution capability but also have detailed error analysis. They also investigate more than 10 models across small to large, closed to open. This will be a valuable resource for readers interested in the capabilities of current LLMs.
- It is interesting to study the "invalid reasoning path," which they define as incorrect intermediate output but correct end results or branch selection, etc. It shows how the model may not follow exactly how to execute the instructions for the current state, unlike a program, and then still get the final answer correct.
- Many other insights are backed by results from many different models. For example, they also investigate the consistency of code execution ability across different test inputs that cover different paths and show that most LLMs are not consistent in the sense that while they can execute some test cases successfully, even with test cases going through the same path, they often may still get them wrong.

**Weaknesses:**

The paper thoroughly investigates many aspects related to the execution path, like strong vs weak reasoning etc. However, it is not clear if the impact of variable values is discussed. For example, it isn't clear how things like large intermediate integers or long lists would affect the CSE results.

**Questions:**

- It is said that valid reasoning is at 83.32% but still only has a low accuracy of 30.79% being correct. Isn't this a bit misleading for the reader before looking into the definition of valid reasoning? The valid reasoning looks like anything but the invalid reasoning, and the invalid reasoning is not defined by whether the intermediate prediction results are wrong. So the valid reasoning containing errors should not be a surprising thing, right?
- Is there a typo in line 362 or line 20 about the number for valid reasoning? (83.32 vs 82.32)

---

> ### Author Response · Authors · 2024-11-27
> **Response to Reviewer QzKa**
>
> **1-** Impact of variable values
>
> Thank you for your suggestion. The revised manuscript contains additional experiments (Appendix A.7) addressing your question. In summary, we show that LLMs struggle the most in predicting variable values of type “**float**” among primitive types. They also struggle more with compound types such as “**list**” than primitive types, which require additional memory and recursion. In Figure 12 of A.7, we observe that LLMs struggle to predict larger integer values and keep track of longer list values.
>
>
>
> **2-** Concerning your review, “valid reasoning is at 83.32% but still only has a low accuracy of 30.79% being correct. Isn't this a bit misleading”
>
> **Starting from the abstract**, we differentiate between the variable “**prediction**” and reasoning “**process**”. The notion of the reasoning process and systematically evaluating it to be valid or invalid is, in fact, the notable contribution of this work. We understand that due to the novelty of the concept and term, it can be confusing at the beginning. Although wordy, we revised the manuscript and added “**process**” when discussing valid or invalid reasoning to avoid confusion.**Please let us know if you have better suggestions.**
>
> **3.-**  **[FLAW]** Concerning your review, “invalid reasoning is not defined by whether the intermediate prediction results are wrong”
>
> Respectfully, the formal definition and evaluation of the invalid reasoning process **indeed considers the intermediate variable value predictions**. The intuition behind formalization is explained in the original version of the paper (Lines 253-263). We further explain them with three cases where the reasoning process is marked as ‘invalid’ by CES:
>
> CASE 1 (Equation 2). The model can correctly predict the output (return value), but at least one of the intermediate predictions is incorrect. During the real execution of the program, the incorrect intermediate state will propagate to the output. Please refer to Figure 2 as an example.
>
> CASE 2 (Equation 3). Regardless of the outcome of output prediction, the model incorrectly predicts the predicate of a conditional statement but correctly predicts the taken branch.
>
> CASE 3 (Equation 4). Regardless of the outcome of output prediction, the model may correctly predict the compound but fails on at least one sub-component. For example, a conditional statement ‘if(x > 0 && y < 0)’ consists of two sub-predicates, ‘x > 0’ and ’y < 0’, if the model correctly predicts the entire statement but fails on one of the two sub-predicates, then the reasoning process is marked as invalid.
>
> **4-** typo in line 362 or line 20
>
> Thanks for your feedback. We have fixed the typo in line 20 in the revised manuscript.

---

### Official Review · Reviewer_ehpE · 2024-11-04

**Soundness:** 3
**Presentation:** 3
**Contribution:** 2
**Rating:** 3
**Confidence:** 4

**Summary:**

This paper introduces Code Execution Simulation (CES), a new benchmark aimed at advancing the evaluation of large language models (LLMs) in code reasoning, particularly for complex programming tasks. CES addresses limitations in existing techniques, which lack comprehensive flow sensitivity and diagnostic capability, by unifying output prediction with key intermediate state evaluations—focusing on flow-sensitive execution paths. CES prompts LLMs at essential decision points (e.g., loop variables, conditions) and leverages adaptive in-context examples for clarity, providing a scalable framework that supports diagnosing reasoning divergence and consistency across varying test coverages. Evaluating thirteen models, including GPT-4 Turbo, Gemini-1.5 Pro, CodeLlama, DeepSeekCoder, Magicoder-S, SemCoder-S, and StarCoder2, on the HumanEval dataset of Python problems, the study finds that while LLMs generally exhibit a high rate of valid reasoning steps (82.32%), their reasoning quality remains predominantly random (55.59%) or weak (41.69%), often falling short in complex flow-sensitive tasks.

**Strengths:**

I think the paper is bringing out an important research question.

The general idea of expecting that LLMs can emulate code if we want to use them for more general software engineering tasks is an interesting one. I would encourage the authors to continue along this research direction.

**Weaknesses:**

Overall I think the paper is not ready for publication yet. The writing of the paper could be improved in places.
For example, in Equation 1 the notation is not clear. CES and GT should have something to differentiate from the variables in the equation. In Figure 7, the radar plot and legend are unclear.

The definition of prime paths being between any two program points requires justification. Could the authors justify this decision. I can imagine that the there are a lot more inputs and dependencies at some intermediate point. An alternative that seems natural would be to consider acyclic paths from the start of a program to some intermediate point. This way the inputs are clearly defined as the inputs to the program.

RQ4 is the most important part of the paper. However, the results are underwhelming currently. The fact that there is no correlation between an LLM correctly emulating a piece of code and the LLM doing well on the programming task for that same piece of code does not support the hypothesis of the paper. Are there other explanations for this observation?

Though I agree with the authors that it would be better if we the LLMs could also emulate the code, I do think this is neither necessary nor sufficient to be able to find bugs, as an example. A lot of humans also find bugs by just pattern matching based on their experience.

I would recommend that the authors explore programs outside of HumanEval, perhaps also exploring other programming languages (C/C++, for instance). The reason being that these programs and programming languages are "too simple" and might require detailed understanding of the program semantics. Perhaps using more complex C/C++ programs involving bitwise operations, pointer arithmetic, etc. and looking at  tasks requiring a more detailed semantic understanding of the program (such as finding security vulnerabilities) might be more conducive to proving the hypothesis of the paper.

**Questions:**

- Why not use paths that always start from the beginning of the program?

- Are there other explanations for RQ4?

- Why restrict to HumanEval programs?

- Did you explore other programming languages other than Python?

---

> ### Author Response · Authors · 2024-11-27
> **Response to Reviewer ehpE**
>
> **1-** Clarifications on Equation 1 and Figure 7
>
> Thanks for the feedback. We updated Figure 7 to make it more clear. Regarding Equation 1, we believe it is sound and complete. The intuition of the formula has also been explained in the draft (through formal definitions in Section 3.1). However, we explained the formula breakdown per individual property for further clarification and help with your re-assessment (Appendix A.5).
>
> **2-** **[FLAW and UNFAIR JUDGEMENT]** Justification on the prime paths and potential alternatives
>
> The prime path is a “**well-known concept**” in program analysis that captures all the execution paths’ properties. Your “**alternative suggestion**” is **incomplete** (implies removing the back edges that are very important in recursive/loop behaviors). Our prime paths also cover what you have suggested (please carefully check the purple box in Figure 5-c). For example, in Figure 5, when the input is ‘max_element([3, 2, -3.5, 0])’, the acyclic paths suggested by you starting from the start are [1,2,3,6] and [1,2,3,4], which are included in the prime paths. When the input is ‘max_element([1,2,3]), the acyclic paths you suggested are [1,2,3,6] and [1,2,3,4,5], which are also included in the prime paths. Prime paths cover more cases than you suggested.
>
> **3-** Explanation of the observations in RQ4
>
> Thanks for your questions in the review. We added three subsections to the appendix with an in-depth analysis of RQ4 observations (Appendix Sections A.5-A.8), which makes the paper stronger. In summary, we demonstrate that frontier LLMs that are more successful in bug-related tasks, indeed, consider code execution simulation in their reasoning process (Figure 9). They are still prone to natural language shortcuts and hallucinations that, despite correct code reasoning, prevent them from correctly performing bug-related tasks (Figure 10). Furthermore, LLMs, when they cannot correctly simulate the execution, still can perform well in bug-related tasks due to natural language shortcuts or out of luck (Figure 11).
>
> **4-** Concerning your review, “it would be better if we the LLMs could also emulate the code, I do think this is neither necessary nor sufficient to be able to find bugs, as an example.”
>
> Our in-depth analysis for RQ4 in Section A.5 (Appendix) shows that frontier models that achieve a higher performance in bug-related tasks (please refer to the first three rows in Table 2), in fact, “**simulate the code execution**” to predict, localize, and repair the bug! This confirms that code execution simulation is certainly “**necessary**” for bug-related tasks. By looking at the cases where models are unsuccessful in CES, but successful in bug-related tasks, we identified the reason to be “incorrect CoT (in natural language) reasonings, shortcuts, and hallucinations.” Please see Figure 11 for examples.
> Please note that we repeated the experiments in RQ4 by modifying the bug-related tasks prompt by including the CoT while prompting. As a result, the numbers in Table 2 have been changed from the original submission.
>
> **5-** **[UNFAIR JUDGEMENT]** Concerning your review, “A lot of humans also find bugs by just pattern matching based on their experience.”
>
> Several highly cited studies of developers show that pattern-based bug finding is “**ineffective**” or even “**unpopular among developers**.” [1] analyzes the reason why developers are not using static analysis tools to find bugs and conclude that false positives can “outweigh” the true positives in volume after intensive user studies. Another recent user study [2]  also points out that understanding the test case is part of developers’ practice in finding and fixing bugs. Therefore, unlike your suggestions, relying on bug patterns is “**insufficient**” for bug findings. Recent research [3,4] has shown that reasoning the runtime behavior can improve LLM’s bug-fixing performance.
>
> **6-** **[UNFAIR JUDGEMENT]** Restricting study on the HumanEval
>
> As explained in the revised draft, one of the most important factors in choosing HumanEval is that it also has a version with human-written injected bugs, which fulfills our design choice in RQ-4. Furthermore, we believe that showing poor code reasoning and hallucinations in a widely used HumanEval benchmark is strong enough to open a new research direction for a more in-depth analysis of Code LLMs' programming results and developing strategies to improve models.
>
> [1] Johnson, B, et al. "Why don't software developers use static analysis tools to find bugs?."
>
> [2] Winter, E., et al. (2022). "How do developers really feel about bug fixing? Directions for automatic program repair."
>
> [3] Ni, A., et al. "NExT: Teaching Large Language Models to Reason about Code Execution."
>
> [4] Ding, Y., et al. "SemCoder: Training Code Language Models with Comprehensive Semantics."

---

### Author Response · Authors · 2024-11-27
**Response to all reviewers**

Dear reviewers, thank you very much for your feedback. We have updated the draft per your comments, and we believe the revised manuscript and our responses should resolve all the concerns you raised. All the additions are highlighted in “**Blue**” in the revised manuscript to make tracking the changes easier.

We identified important “**flaws**” in some reviews, which have been used to question the “**soundness**” of this work “**unfairly**.” Respectfully, we ask the reviewers to read our responses and the updated manuscript. We would appreciate your revising their reviews, assessments, and scores accordingly. We also want to ask for consistency between your review and scores. **All the reviewers consider this work novel**, and we have addressed all other concerns. Rejecting a novel work with important contributions should be based on essential flaws in the work, which we believe do not hold.

We would like to highlight an important fact that has yet to be considered crucial in evaluating the contributions of this work: Regardless of your point of view, code reasoning and, specifically, execution reasoning is becoming an important evaluation criterion in Code LLMs. This work **raises a concern about the validity** of the code reasoning results using existing approaches. It also proposes **a systematic approach** to rule out the model’s hallucinations on execution reasoning to ensure a valid and proper evaluation of LLM’s code reasoning. To the best of our knowledge, as active researchers in LLM reasoning and evaluation, a systematic approach for identifying LLM’s hallucinations and ruling them out can be considered **a remarkable contribution**. We hope the revised manuscripts with additional experiments and analysis resolve your other concerns and that your re-assessment considers this important contribution.

---

### Note · Authors · 2025-03-12

I have read and agree with the venue's withdrawal policy on behalf of myself and my co-authors.

---

### Meta-Review · Area_Chair_AaVZ · 2024-12-18

**Metareview:**

Claims and findings: A new measure (CES) of code reasoning ability, and analysis suggesting that LLMs leverage more pattern matching abilities for program synthesis compared to general reasoning faculties that allow them to execute code.

Strength: Scientific investigation of LLM abilities is valuable to the community as opposed to merely cheerleading LLMs. The proposed analysis is novel.

Weaknesses: relatively limited evaluation (HumanEval), making it unclear if the empirical results are a consequence of the dataset or a more general phenomenon; lack of general insight on why the proposed CES has little correlation with other coding tasks.

Reason for rejection:  Although the reviewers (and the area chair) think that the paper poses intriguing questions, it does not dive deep enough on the empirical side (confining itself to only one simple benchmark) nor on the theoretical side (lacking deep insight on why these phenomena occur). At least one should be present.

**Additional Comments On Reviewer Discussion:**

The reviewers raised the limited evaluation, which was not adequately rebutted. There was revision to help give more analysis, but absent a broader empirical evaluation it is unclear what value that analysis contributes. Last, the rebuttal was surprisingly combative: In the future would advise the authors that they will have better luck persuading reviewers with short punchy rebuttals than criticizing the character rather than the content of the review.

---

### Decision · Program_Chairs · 2025-01-22

Reject